



# PyRTlib: an educational Python-based library for non-scattering atmospheric microwave Radiative Transfer computations

Salvatore Larosa[1], Domenico Cimini[1,2], Donatello Gallucci[1], Saverio Teodosio Nilo[1], and Filomena
Romano[1]
[1]National Research Council of Italy, Institute of Methodologies for Environmental Analysis (CNR-IMAA), Tito Scalo
(Potenza), 85050, Italy
[2]CETEMPS, University of L'Aquila, L'Aquila, 67100, Italy
*Correspondence to*: Salvatore Larosa (salvatore-larosa@cnr.it)
**Abstract.** This article introduces PyRTlib, a new standalone Python package for non-scattering line-by-line microwave
Radiative Transfer simulations. PyRTlib is a flexible and user-friendly tool for computing down and up-welling brightness
temperatures and related quantities (e.g., atmospheric absorption, optical depth, opacity, mean radiating temperature) written
in Python, a language commonly used nowadays for scientific software development especially by students and early career
scientists. PyRTlib allows simulating observations from ground-based, airborne, and satellite microwave sensors in clear sky
and in cloudy conditions (under non-scattering Rayleigh approximation). The intention for PyRTlib is not to be a competitor
for state-of-the-art atmospheric radiative transfer codes that excel for speed and/or versatility (e.g., ARTS, RTTOV). The
intention is to provide an educational tool, completely written in Python, to readily simulate atmospheric microwave radiative
transfer from a variety of input profiles, including predefined climatologies, global radiosonde archives, and model reanalysis.
The paper presents quick examples for the built in modules to access popular open data archives. The paper also presents
examples for computing simulated brightness temperature for different platforms (ground-based, airborne, and satellite), using
various input profiles, showing how to easily modify other relevant parameters, such as observing angle (zenith, nadir, slant),
surface emissivity, and gas absorption model. PyRTlib can be easily embedded in other Python codes needing atmospheric
microwave radiative transfer (e.g., surface emissivity models and retrievals). Despite its simplicity, PyRTlib can be readily
used to produce present-day scientific results, as demonstrated by two examples showing (*i*) absorption model comparison and
validation with ground-based radiometric observations and (*ii*) uncertainty propagation of spectroscopic parameters through
the radiative transfer calculations following a rigorous approach. To our knowledge, the uncertainty estimate is not provided
by any other currently available microwave radiative transfer code, making PyRTlib unique for this aspect in the atmospheric
microwave radiative transfer code scenario.



## 1 Introduction

Radiative transfer (RT) models play a fundamental role in atmospheric sciences, as they are broadly used to simulate how electromagnetic radiation travels through the atmosphere as it interacts with atmospheric constituents (such as gases, aerosols and hydrometeors) through absorption, emission, scattering, and refraction. RT models are commonly used as forward operators to simulate and understand remote sensing observations from any platform, ground-based, airborne or spaceborne. RT calculations depend on the state of the atmosphere (pressure, temperature, composition), the optical properties of the atmospheric constituents (molecules and particles), the simulated observing geometry, and the spectral range. Given a set of specifications on spectral range, atmospheric conditions, and observing geometry, the RT model is able to compute the atmospheric opacity and the observations simulated accordingly. Simulated observations are then used in a broad range of applications, from atmospheric process understanding, the retrieval of atmospheric variables, to data assimilation into Numerical Weather Prediction (NWP) models. Although the theoretical aspects of wave-atmosphere interactions are essentially the same throughout the electromagnetic spectrum, different RT models have been developed to account for the specific features of limited spectral ranges, such as the visible, infrared, and microwave portions of the electromagnetic spectrum. In particular, several microwave (MW) RT models have been developed throughout the years to serve the needs of passive remote sensing from MW radiometers (e.g., Liebe, 1989; Buehler et al., 2005; Rosenkranz, 2017). Many examples are available in the open literature on the use of MW RT models for atmospheric sciences, including but not limited to: process understanding (Tripoli et al., 2005; Martinet et al 2017), atmospheric retrieval development (Eriksson et al., 2005; Boukabara et al., 2013; Sanò et al., 2015; Larosa et al., 2023), MW instrument design and validation (Buehler et al., 2012; Fox et al., 2017), data assimilation into NWP model (Eyre et al., 2020; Martinet et al., 2020), instrument synergy (Marzano et al., 1999; Turner and Löhnert, 2021; Cimini et al., 2023). A variety of software codes has been developed throughout the past three decades for implementing different flavours of available MW RT models, differing for features, assumptions, approximations, as well as coding languages. Among the different features, RT codes may be classified in scattering or non-scattering (i.e., considering absorption only). Similarly, RT codes may be classified as line-by-line, meaning that RT can be modelled at any frequency from the contributions of many gas absorption lines, or parameterized, meaning that RT can be modelled at a limited number of channels for which the optical depth is parameterized considering their spectral response function, initially trained with line-by-line calculations. Other assumptions include the observing geometry, going from plane-parallel one dimensional (1-D) calculations that consider the atmosphere state changing only in the vertical dimension, to higher dimensional (2-D or 3-D) geometries, which allow to consider also the horizontal spatial dishomogeneity, to spherical geometry, allowing to properly model the atmospheric shape and its effect on the bending angle of the radiation path. Although RT codes enabling line-by-line, scattering, and spherical geometry computations are much more complex and computational demanding than the parametrized, non-scattering, and 1-D plane-parallel assumptions, they allow more accurate modelling of the impact of spectral resolution, particle size, and 3-D distribution, respectively. Concerning the coding language, most of MW RT software codes are available in compiled programming languages such as C, C++ and Fortran. However, the interpreted programming



language Python has become increasingly popular for scientific computing in the last decades, thanks to its numerous extension
packages, and it is now widely considered the language of choice in many areas, including atmospheric science. Therefore,
some of the available RT codes allow users to access their features by running Python modules as wrapper of the core software,
although the core software needs to be compiled from source or in binary form to access such modules. There are also cases
for which the original code has been translated into Python. Table 1.1 reports a list of most popular MW RT codes, by no
means complete, with their key features and access information. In the following, a brief introduction is given of the most
relevant MW RT codes for this paper.

**Table 1.1:** List of popular codes suitable for atmospheric radiative transfer in the microwave spectral region.

| Name | References | line-by-line/band | Scattering | Language | License | Access |
|---|---|---|---|---|---|---|
| ARTS | Eriksson et al., 2011; Buehler et al., 2018 | line-by-line | Yes | C++ (python interface as wrapper) | GPL v3 | https://www.radiativetransfer.org/ |
| CRTM | Han et al., 2006; Ding et al., 2011; Wei et al., 2022 | line-by-line/band | Yes | Fortran (python interface as wrapper) | CC0 v1.0 | https://www.jcsda.org/jcsda-project-community-radiative-transfer-model |
| MonoRTM | Clough et al., 2005 | line-by-line | Yes | Fortran | GPL | http://rtweb.aer.com/monortm_frame.html |
| PAMTRA | Mech et al., 2020 | line-by-line | Yes | Fortran (python interface as wrapper) | GPL v3 | https://pamtra.readthedocs.io/ |
| Py4CAtS | Schreier et al, 2019 | line-by-line | No | Python | GPL | https://atmos.eoc.dlr.de/tools/Py4CAtS/ |
| RTTOV | Saunders et al., 2018 | band | Yes | Fortran (python interface as wrapper) | available on request | https://nwp-saf.eumetsat.int/site/software/rttov/ |
| RTTOV-gb | De Angelis et al., 2016 Cimini et al., 2019 | line-by-line and band | Yes | Fortran | available on request | https://nwp-saf.eumetsat.int/site/software/rttov-gb/ |
| TBUPDN | Rosenkranz, 2017 | line-by-line and band | No | Fortran | freely available | http://cetemps.aquila.infn.it/rttovgb/lblmrt_ns.html |


**ARTS**: The Atmospheric Radiative Transfer Simulator (ARTS) is a radiative transfer model suitable for calculations from the
microwave to the thermal infrared spectral range (Buehler et al., 2005; Eriksson et al., 2011; Buehler et al., 2018). ARTS is



implemented in C++ with a modular design, allowing the flexibility for performing many different applications concerning
radiative transfer calculations in all viewing geometries from inside or outside the atmosphere: uplooking, downlooking, limb-
looking. ARTS allows the choice of different state of the art absorption models, including line-by-line from HITRAN or other
catalogues plus various absorption continuum parameterizations. It is fully polarized, allowing RT calculations from 1 to 4
Stokes components. It allows scattering computations from spherical and non-spherical atmospheric particles. It also provides
analytical or semi-analytical Jacobians for a large set of state parameters. It supports XML and NetCDF file format for data
import and export. ARTS can be run standalone or through external tools, such as PyARTS, a Python package that serves as
wrapper for the main ARTS core library. PyARTS is part of the ARTS source repository. PyARTS provides an interactive
interface to the ARTS engine for running radiative transfer simulations and has many ARTS built-in types for the manipulation
of input data and the evaluation simulation results. However, PyARTS cannot be run as a standalone python package as it
needs ARTS built before.

**CRTM**: The Community Radiative Transfer Model (CRTM; Han et al., 2006; Ding et al., 2011; Wei et al., 2022) is a fast
radiative transfer model developed to efficiently simulate specific spaceborne Earth observing sensors. The CRTM was
developed by the U.S. Joint Center for Satellite Data Assimilation (JCSDA) to be a library for users to link to from other
models. However, CRTM can be run in "stand-alone" mode. CRTM is a sensor-based RT model, supporting more than 100
sensors on meteorological and other remote sensing satellites, covering wavelengths ranging from the visible through the
microwave. The source code is written in standard Fortran95 and makes extensive use of modules and derived type data
structures. CRTM includes both the forward model and its Jacobian with respect to the input atmospheric state variables,
accounting for the absorption of atmospheric gases as well as the multiple scattering of water and ice clouds composed of
spherical and a variety of nonspherical particles, working under all atmospheric and surface conditions. CRTM is extensively
used in several applications, such as the NOAA Microwave Integrated Retrieval System (MiRS), the NCEP data assimilation
system, and the NOAA STAR Integrated Calibration/Validation System Long-Term Monitoring System. CRTM can be called
from Python scripts using pyCRTM, which embeds CRTM Fortran data structures and procedures directly into Python, taking
advantage of both the simplicity and ease of use of Python syntax and the flexibility that comes from the extensive Python
ecosystem (Karpowicz et al., 2022).

**MonoRTM**: MonoRTM represents an atmospheric radiative transfer model widely used in the scientific community to
generate simulated spectral radiance ranging from the ultraviolet to the microwave region (Clough et al, 2005). It has been
produced by the Atmospheric & Environmental Research (AER) and is based on the same physical properties and continuum
absorption model as the Line-By-Line Radiative Transfer Model (LBLRTM), which is also developed and maintained by AER.
These are both Fortran 90 codes, however MonoRTM is particularly suitable to simulate a single or a set of few monochromatic
wavelengths. Atmospheric molecular absorption covers all spectral regions, with molecular optical depths computed within
the Monochromatic Optical Depth Model module; however, spectral radiance calculation in the presence of cloud liquid water





is only possible in the microwave range and relies on the model developed by Liebe et al. (1991). MonoRTM also accounts
for molecular absorption within the spectral line center, by using the MT_CKD continuum (Clough et al., 2005). Line coupling
effects, which are crucial for e.g., oxygen lines in the microwave region, are also dealt with in the code (Rosenkranz, 1988;
Tretyakov et al., 2005; Cadeddu et al., 2007).
**PAMTRA**: This is an atmospheric radiative transfer code, namely the Passive and Active Microwave radiative TRAnsfer
(PAMTRA, Mech et al., 2020), specifically designed to simulate both passive microwave radiances as well as active remote
sensing measurements in the presence of cloudy atmosphere. PAMTRAM exploits the passive forward model to compute both
upward and downward looking polarized brightness temperatures and radiances; regarding radar measurements instead, the
active forward model yields Doppler spectra and relative moments, e.g. reflectivity, mean Doppler velocity, skewness, and
kurtosis. The model is built within a Fortran-Python environment, allowing the flexibility to different input/output formats and
instrument characteristics (e.g., observations from ground-based, airborne or spaceborne platforms, viewing angles, etc..), with
the assumption of a plane-parallel, one-dimensional homogeneous atmosphere over the horizontal direction. The user can
select several operational modes among scattering and absorption models, within a wide range of spectroscopic parameters
and databases; the absorption unit is based on the Millimeter-wave Propagation Model (MPM; Liebe, 1989). Generally,
pyPAMTRA is used in the scientific community, which features a Python wrapper built around the Fortran core, allowing
direct access from Python, without using the I/O Fortran routines. The pyPAMTRA interface makes the model user-friendly,
simplifying the importing of model data, the output in terms of files or plots, and the parallel running of the code on a multicore
processor or cluster machines.
**Py4CAtS**: Python scripts for Computational Atmospheric Spectroscopy (Py4CAtS, Schreier et al., 2019), is a software
designed for computing atmospheric spectroscopy both in the infrared and microwave spectral regions. It was initially
conceived to enable Python access to a previous Fortran 90 Generic Atmospheric Radiation Line-by-line Code (GARLIC,
Schreier et al., 2014). Later on, it has become a complete self-consistent independent software, based entirely on Python
numerical array processing modules, providing line by line radiances, as well as absorption cross sections and coefficients,
optical depths, transmissions and weighting functions.  Py4CAtS consists of a set of modules and functions allowing to
generate line-by-line cross sections for given pressure(s) and temperature(s), to combine cross sections into absorption
coefficients and optical depths, and to integrate along the line-of-sight into transmission and radiance/intensity. Py4CAtS is
also user-friendly, since it offers an interactive environment and the possibility to perform batch line-by-line modeling. The
software can be started within the console terminal, the Python interpreter or the Jupyter Notebook; besides, all intermediate
variables can be visualized too. Py4CAtS relies on a plane-parallel atmosphere assumption, and considers non-scattering
interactions, with the Schwarzchild equation featuring thermal emission as source only; furthermore, neither continuum nor
collision-induced absorptions are taken into account as contributions to the molecular absorption, which is therefore limited to
the Voigt Line shape.




**RTTOV**: Similar to CRTM, the Radiative Transfer for TOVS (RTTOV) is a fast radiative transfer model for modelling passive
visible, infrared and microwave downward-viewing satellite radiometers, spectrometers and interferometers (Saunders et al.,
2018; Hocking et al., 2021). RTTOV is a FORTRAN 90 code designed to be incorporated within user applications for
simulating satellite radiances. RTTOV is developed and maintained by the NWP Satellite Application Facility of EUMETSAT,
and it is probably the most used RT code for satellite data assimilation into NWP models. Given an atmospheric profile of
temperature, water vapour and, optionally, trace gases, aerosols and hydrometeors, together with surface parameters and a
viewing geometry, RTTOV computes the top of atmosphere radiances for a set of space-borne sensors from past, current, and
future satellite Earth observing missions. The core of RTTOV is a fast parameterisation of layer optical depths due to gas and
liquid water absorption. Profiles of layer-to-space transmittances computed by the line-by-line code AMSUTRAN (Turner et
al., 2019) are the basis for the training of the fast parameterisation. RTTOV consists in both the forward model, which simulates
the upwelling radiances for a given sensor, and its Jacobian, which calculates the radiance derivatives with respect to the input
atmospheric state variables. RTTOV includes scattering calculations for simulating cloudy and aerosol-affected radiances in
the infrared. Scattering at MW frequencies from hydrometeors of different phases and shapes is available through the wrapper
code RTTOV-SCAT (Bauer et al., 2010; Geer et al., 2017). RTTOV has a built-in graphic user interface (GUI) which allows
the user to modify an atmospheric/surface profile, run RTTOV for a given instrument, produce radiances and brightness
temperatures, calculate Jacobians, perform a basic retrieval, and display instantaneously the results. RTTOV is natively in
Fortran, but Python wrappers are available to allow the functionality of RTTOV in Python. These wrappers provide Python
bindings for the RTTOV Fortran code, making it easier for Python users to use. A ground-based version of RTTOV for
simulating ground-based MW sensors is also available, though limited to version 11 (De Angelis et al., 2016; Cimini et al.,
162   2019).


**TBUPDN**: The upward-downward Tb (TBUPDN) code is a library of Fortran routines for the non-scattering line-by-line
microwave RT simulations (Rosenkranz, 2017). The code is developed and maintained by Philip W. Rosenkranz since more
than 30 years (Rosenkranz, 1993). TBUPDN is intended as an educational tool with limited ranges of applicability, i.e.
calculations of upward- and downward-propagating $T_B$ respectively at the top and bottom of the atmosphere. The main routines
can be run stand-alone or read as examples for using the subroutines (e.g., the absorption model routines) in other software
programs. A major feature of TBUPDN is the continuous update of absorption routines, originally based on the MPM code,
with subsequent spectroscopic modifications from most recent findings from laboratory and field campaign experiments
(Rosenkranz, 1988; 1998; 2001; 2005; Rosenkranz and Cimini, 2019; Gallucci et al., 2023). User interfaces are provided for
handling I/O text files and produce encapsulated postscript figures.

Table 1 and the list above are meant to provide an overview of open access codes that are used extensively by the MW
community, but do not pretend to be complete. Other codes suitable for atmospheric RT in the MW are available, either openly



or commercially, e.g., BTRAM (Chapman et al., 2010), MODTRAN (Berk et al., 2014). Other RT codes that are available in
Python or with a Python interface, although not concerning MW in Earth's atmosphere, include the following: PYDOME
(Efremenko et al. 2019) for simulating satellite measurements of reflected and scattered solar radiation in the ultraviolet and
visible spectral ranges; Py6S (Wilson, 2013), a Python interface to the 6S RTM (Vermote et al., 1997) designed to simulate
solar radiation through atmospheres on Earth and other planets; PySMARTS module (Ayala Pelaez and Deline, 2020) contains
functions for calling SMARTS: Simple Model of the Atmospheric Radiative Transfer of Sunshine to compute clear sky spectral
irradiances (on a tilted or horizontal receiver plane) for specified atmospheric conditions; petitRADTRANS (Mollière et al.,
2019) and PYRATE (Tritsis et al., 2018) for simulating RT through atmospheres on exoplanets.

This paper introduces PyRTlib, a new standalone Python package for non-scattering line-by-line microwave RT simulations.
Given the premises above, one may ask: is a new RT code really needed? The intention for PyRTlib is not to be considered a
competitor for the codes mentioned above, which represent the cutting edge with their own peculiarities, in terms of efficiency,
flexibility, modularity, and applicability. Nevertheless, the reasons behind the development of PyRTlib are the following:
1)   Develop an educational tool, similarly to TBUPDN, but in Python, which represents nowadays the most used language
190        for scientific software development, especially by students and younger scientists;
2)   Provide user-friendly Python interfaces, similarly to PyARTS or pyPAMTRA, to compute MW RT simulations using
192        popular datasets as input, such as radiosonde repository or global reanalysis;
3)   Allow easy comparison of MW calculations using different atmospheric absorption models, e.g., those proposed
194        throughout the last three decades, for any platform (ground-based, airborne, and spaceborne) and observing geometry
195        (zenith, nadir, slant);
4)   Provide $T_B$ calculations with the associated uncertainty due to the uncertainty on spectroscopic parameters, following
197        a general rigorous approach recently outlined (Cimini et al., 2018; Gallucci et al., 2023).


In particular, to our knowledge the uncertainty estimate is not available by any other MW RT code, making PyRTlib unique
for this aspect in the MW RT scenario. Thus, this paper provides a description of PyRTlib version 1.0 and advocates its use
through a range of examples demonstrating its value in producing passive MW simulations from notable input datasets
(radiosondes, reanalysis) and for ground-based, airborne, and satellite perspectives.
The paper is structured as follows: brief introduction of the basics of equations of radiative transfer model, the main absorption
model available and how profiles can be interpolated and extrapolated are discussed in section 2. The tools for retrieving and
managing input data from open access repositories (e.g., radiosonde observations and model reanalysis) are discussed in
Section 3. Usage of the code as well as some implementation details and a few examples of applications are presented in
Section 4. Section 5 summarises the conclusions and future developments, while Section 6 provides instructions for code
availability and usage.



## 2 Radiative transfer model

An atmospheric RT model simulates the propagation of electromagnetic radiation through the atmosphere as it interacts with the atmospheric constituents (gases, aerosols and hydrometers) through absorption, emission, scattering, and refraction. The intensity of radiation $I$, also called radiance, expresses the power carried by the electromagnetic radiation along the direction of propagation per unit area and solid angle at a given frequency $f$. Considering an ideal blackbody radiator in local thermodynamic equilibrium at physical temperature $T$, the intensity of radiation $I$ is given by the Planck function:

$$B_f(T) = \left(\frac{2hf^3}{c^2}\right)\left(\frac{1}{e^{\frac{hf}{kT}}-1}\right)$$
(1)

where h and k are the Planck and Boltzmann constants, respectively, and c is the speed of light. From Eq. (1) comes directly the definition of brightness temperature $T_B$, as the temperature that a blackbody radiator should have to emit the radiance $I$, i.e., $I = B_f(T_B)$.

The relevance of radiation scattering by atmospheric particles depends on the ratio between the size of the scattering particle $r$ and the radiation wavelength $\lambda$, so called size ratio $x = 2\pi r/\lambda = 2\pi rf/c$ (Petty, 2006). If $x \ll 1$, then the contribution of scattering can be considered negligible. That is the case at microwave and millimeter-wave frequencies ($\lesssim 1$ THz) in clear sky (no clouds). For relatively small hydrometeors (i.e., liquid and ice clouds) the size ratio is still $x < 1$, and the Rayleigh approximation is valid, for which absorption is still dominant with respect to scattering. Thus, a simplifying common assumption at microwave frequencies is to neglect atmospheric scattering, which is commonly assumed valid in absence of large particles (i.e., liquid and solid precipitation). In such a case, the Swartzchild equation applies, i.e.:

$$B_f(T_B) = B_f(T_{BG})e^{-\tau_f(0,\infty)} + \int_0^\infty B_f(T(s))\alpha_f(s)e^{-\tau_f(0,s)}ds$$
(2)

where s indicates the position along the propagation direction, $\alpha_f$ indicates the atmospheric absorption coefficient, $\tau_f$ indicates the atmospheric opacity ($\tau_f(a,b) = \int_a^b \alpha_f(s)ds$), and the two extremes of the integral indicate the position where the $T_B$ measurement is taken (0) and the position of a uniform background ($\infty$) of temperature $T_{BG}$. The first term of Eq.(2) changes depending on the observing geometry. For an uplooking radiometer measuring downwelling radiation, without discrete sources (such as the Sun or Moon) within the antenna field of view, $B_f(T_{BG})$ is simply equal to $B_f(T_{CBG})$, where $T_{CBG} \simeq 2.7\ K$ is the microwave cosmic background brightness temperature (Rosenkranz, 1993). For a downlooking radiometer measuring upwelling radiation, e.g. from a satellite platform, a typical background is the Earth's surface, the spectral emissivity ($\varepsilon_f$) of which must be taken into account to model the complementary contribution of Earth's surface emission and reflection of





downwelling radiation. Thus, indicating with SRF the position of the Earth's surface and TOA the top of atmosphere, $B_f(T_{BG})$
in Eq. (2) becomes:

$$B_f(T_{BG}) = (1 - \varepsilon_f)\left[B_f(T_{CBG})e^{-\tau_f(SRF,TOA)} + \int_{SRF}^{TOA} B_f(T(s))\alpha_f(s)e^{-\tau_f(SRF,s)}ds\right] + \varepsilon_f B_f(T_{SRF}) \qquad (3)$$

The integral in the atmospheric terms in Eq. (2) and (3) is divided into the sum of integrals over each of the NL-1 layers in
between the NL levels in which the atmosphere is discretized (Schroeder and Westwater, 1991). In case of uplooking
simulations of downwelling radiation:

$$\int_0^\infty B_f(T(s))\alpha_f(s)e^{-\tau_f(0,s)}ds = \sum_{i=2}^{NL}\int_{s_{i-1}}^{s_i} B_f(T(s))\alpha_f(s)e^{-\tau_f(0,s)}ds \qquad (4)$$

The integrals in the second term can be simplified by introducing a mean radiating temperature of a layer $T_{MR}$, such as:

$$\int_{s_{i-1}}^{s_i} B_f(T(s))\alpha_f(s)e^{-\tau_f(0,s)}ds = B_f(T_{MR})\int_{s_{i-1}}^{s_i}\alpha_f(s)e^{-\tau_f(0,s)}ds = B_f(T_{MR})e^{-\tau_f(0,s_{i+1})}\left[1 - e^{-\tau_f(s_{i+1},s_i)}\right] \qquad (5)$$

$B_f(T_{MR})$ can be approximated as the weighted average of $B_f(T(s))$ from the two profile levels that form the layer:

$$B_f(T_{MR}) \simeq \frac{B_f(T(s_{i-1})) + B_f(T(s_i))e^{-\tau_f(s_{i-1},s_i)}}{1 + e^{-\tau_f(s_{i-1},s_i)}} \qquad (6)$$

where the exponential weight $e^{-\tau_f(s_{i-1},s_i)}$ represents the attenuation of $B_f(T(s_i))$ over the layer between levels $i$ and $i$-$1$. The
case of downlooking observations can be simply derived from the above. The contribution of each layer is then summed up as
in Eq. (4) (Schroeder and Westwater, 1991).

**2.1 Modelling atmospheric absorption**
Modelling atmospheric absorption is a crucial component of RT codes. Absorption models are based on parameterized
equations to calculate atmospheric absorption ($\alpha_f$ in Eq. (2)) given the constituents' concentration and their thermodynamic
conditions (Rosenkranz, 1993). Note that, as introduced earlier, PyRTlib is a non-scattering RT code, i.e., it assumes that
attenuation is due entirely to absorption by atmospheric gases and cloud water, while it neglects the extinction due to particle
scattering. Concerning atmospheric gases, PyRTlib considers the absorption contribution by nitrogen and oxygen (also called
dry air contribution, $\alpha_{dry}$) and water vapour (wet contribution, $\alpha_{wet}$). These three species sum up to more than 99% of the



atmospheric gas mixture and account for most of the gas absorption in the MW spectrum. PyRTlib also offers the option to
add the contribution of ozone ($\alpha_{O3}$); this causes a relatively small absorption increase in very narrow spectral ranges due to
many nearly monochromatic spectral lines at the expense of slower computations. Concerning hydrometeors, the absorption
of cloud liquid ($\alpha_{liq}$) and ice ($\alpha_{ice}$) particles are considered. Note that $\alpha_{dry}, \alpha_{wet}, \alpha_{O3}, \alpha_{liq}$, and $\alpha_{ice}$ all depend on frequency
and location in space, although not shown for simplicity. In fact, the sum $\alpha_{dry} + \alpha_{wet} + \alpha_{O3} + \alpha_{liq} + \alpha_{ice}$ represents $\alpha_f(s)$ in
Equations (2) and (3). Of course the terms $\alpha_{liq}$ and $\alpha_{ice}$ are zero in clear sky conditions while $\alpha_{O3}$ is zero if ozone contribution
is neglected. Absorption models for computing $\alpha_{dry}, \alpha_{wet}, \alpha_{O3}, \alpha_{liq}$, and $\alpha_{ice}$ from the constituents' concentration and the
thermodynamic conditions are available in the open literature (e.g., Rosenkranz, 1993). These models rely on parameterized
equations and spectroscopic parameters, valid up to 1 THz, determined through theoretical calculations and/or laboratory and
field measurements. These settings are continuously updated and improved (Liebe et al., 1989; Rosenkranz, 1998; Liljegren
et al., 2005; Turner et al., 2009; Mlawer et al., 2012; Koshelev et al., 2018). The proposed changes are occasionally summarised
in review articles (e.g., Rothman et al., 2005; Gordon et al., 2017, Rosenkranz, 1998; 2017; Tretyakov, 2016). In particular,
PyRTlib implements absorption routines originally based on the MPM code (Liebe, 1989), with subsequent spectroscopic
modifications from laboratory and field campaign experiments (Rosenkranz, 1988; 1998; 2005; 2015; 2017; Rosenkranz and
Cimini, 2019; Koshelev et al., 2021; 2022). These changes have been summarised in two papers (Cimini et al., 2018; Gallucci
et al., 2023). PyRTlib provides the possibility to easily compare different absorption model configurations, as discussed in one
of the examples in Section 4. In case the input profile contains non-zero cloud liquid and/or ice water, the relative absorption
is computed and added to the total absorption. The cloud absorption model used here assumes Rayleigh approximation, under
which scattering is negligible relative to absorption, and absorption is independent of cloud particle size distribution. These
assumptions restrict the model to non-precipitating clouds with particle radii less than about 100 $\mu$m for frequency less than
100 GHz. Therefore, in its current version, PyRTLib is not adequate for modelling extinction by rain or large cloud droplets
or ice particles. Absorption by cloud liquid ($\alpha_{liq}$) and ice ($\alpha_{ice}$) particles are implemented following the algorithms described
in Schroeder & Westwater (1991), and later improvements (i.e., Liebe et al., 1991; 1993). Optionally, a model designed
specifically for the absorption of supercooled liquid water particles (Rosenkranz, 2015) is also implemented.

## 2.2 Modelling a continuous atmosphere

To compute absorption throughout the atmosphere, the gas concentrations and thermodynamic profiles are to be provided in
input. While $O_2$ concentration is assumed constant with altitude, concentration of $H_2O$ (and $O_3$, if considered) is usually
variable with altitude, and similarly for air pressure and temperature. These inputs may come from atmospheric measurements
(e.g., balloon-borne radiosoundings) or atmospheric model output (e.g., NWP model), and are typically available at discrete
levels. To compute realistic simulated observations from ground-based or satellite platforms, the profiles must cover the
vertical range from the Earth surface to a reasonable top-of-atmosphere (TOA), where the atmosphere is so rare that it affects





MW radiation negligibly. In practice, the TOA is assumed to approximate the infinite limits on the integral in Eq.(2). For
simulations of downwelling radiation, the vertical range could be reduced at the bottom to the height of the simulated receiving
antenna (e.g., for the simulations of airborne or elevated instruments). PyRTlib provides functions to extend the input profile
to a TOA at 0.1 mb (following Schroeder and Westwater, 1991), a pressure well below the minimum pressure (i.e., maximum
altitude) reached by radiosoundings. This profile extension follows a recommendation (ITU-R P.835-6, 2017) by the
International Telecommunication Union - Radiocommunication Sector (ITU-R), providing expressions and data for reference
standard atmospheres required for the calculation of gaseous attenuation on Earth-space paths. In particular, PyRTlib currently
implements the data in Annex 1, i.e., standard atmospheres to be used to determine temperature, pressure and water-vapour
pressure as a function of altitude, when more reliable local data are not available. Data in Annex 3, i.e. providing vertical
profiles capturing diurnal, monthly, and seasonal variations from ECMWF 15-year data set re-analysis (ERA15) will be
implemented in future PyRTlib releases. Another option is to increase the level density by adding levels through interpolation.
This option allows a maximum pressure difference between a pair of adjacent profile levels. If the pressure difference in the
input profile exceeds the specified maximum value, PyRTLib divides the layer between the two levels into the smallest number
of equally-spaced pressure levels that differ by less than the specified maximum value, using linear interpolation in natural
logarithm of pressure.

### 2.3 Modelling observation geometry

The input height profile $h$ is assumed to represent the vertical line-of-sight ray path coordinate. This corresponds to $s$ in Eq.(2)
for uplooking zenith-pointing simulations and to $h_{TOA}$-$s$ for downlooking nadir-pointing simulations. For observing angles
different from zenith or nadir, the ray path increases due to the slant path through the atmosphere. Considering a plane-parallel
atmosphere, the increase effectively corresponds to the multiplicative factor $secant\phi$, where $\phi$ is angle with respect to
zenith/nadir (or $cosecant\theta$, if the elevation angle $\theta$ is considered). This approximation is the default option in PyRTLib.
Atmospheric refraction can also be considered, which affects the ray path by radiation bending. Following Schroeder and
Westwater (1991), the ray path is modelled assuming a spherically stratified atmosphere for which the radio wave path obeys
Snell's law (Schroeder and Westwater, 1991):

$n\,r\,cos\theta = constant$            (7)

where $n$ is the atmospheric refractive index and $r$ is the radial distance from the center of the Earth to a point on the ray path.
All these qualities depend on height above the surface. The refractive index $n$ is computed from the dry and wet refractivity
($N_d$ and $N_w$, respectively) and the inverse compressibility of dry air and water vapor ($Z_d^{-1}$ and $Z_w^{-1}$, respectively) through the
following non-dispersive model:



$\quad n = 1 + (N_d + N_w) \cdot 10^{-6}$ (8)
$\quad N_d = k_1 \left(\frac{P_d}{T}\right) Z_d^{-1}$ (9)
$\quad N_w = \left(k_2 \frac{e}{T} + k_3 \frac{e}{T^2}\right) Z_w^{-1}$ (10)
$\quad Z_d^{-1} = 1 + P_d \left(57.90 \cdot 10^{-8} \left(1 + \frac{0.52}{T}\right) - 9.4611 \cdot 10^{-4} \frac{T-273.16}{T^2}\right)$ (11)
$\quad Z_w^{-1} = 1 + 1650 \frac{e}{T^3} (1 - 0.01317 \, T_C + 1.75 \cdot 10^{-4} \, T_C^2 + 1.44 \cdot 10^{-6} \, T_C^3)$ (12)

where $T, e, P_d$ are respectively the air temperature (K), water vapour partial pressure, and dry air partial pressure (hPa), while
$T_C$ is the air temperature in °C ($T_C = T - 273.16$). The three $k$ coefficients ($k_1 = 77.604, k_2 = 64.79, k_3 = 3.776 \cdot 10^5$) are
given by Saastamoinen, 1972 and references therein. PyRTlib optionally provides a slightly modified definition for computing
the dry and wet refractivity terms, though leaving the total refractivity and the refractive index unaffected, which is commonly
used in geodesy (ESA TN, 2019). Finally, for each specified observing angle a ray-tracing algorithm based on Eq.(7) is used
to compute the refracted path length between each pair of adjacent profile levels. The integrals along the ray path are computed
assuming that the integrand variable decays exponentially with height within the layer defined by a pair of adjacent levels.
With this assumption, the integral along one layer of a general integrand variable $X$ is given by (Schroeder and Westwater,

348 1991):


$\quad \int_{i-1}^{i} X(s)ds = (s_i - s_{i-1})[(X(s_i) - X(s_{i-1})) \, / \, ln(X(s_i)/X(s_{i-1}))]$ (13)

This is used to compute path-integrated quantities, such as layer-integrated absorption profiles for RT calculations as well as
total integrals along the entire path, such as precipitable water vapour, path delay (excess path length) due to dry air and to
water vapour, and total absorption due to water vapour, dry air, and cloud liquid/ice.
**3 Tools for retrieving input data**
PyRTlib comes with a built-in module to easily retrieve meteorological data that can be used as input for the RT calculations.
These modules allow easy access to data repositories of radiosonde observations (RAOB) and model reanalysis. The RAOB
repositories currently considered in PyRTlib are the University of Wyoming Upper Air Archive (UWYO, 2015) and the U.S.
National Ccenter for Environmental Information (NCEI) Integrated Radiosonde Archive (IGRA) version 2 (Durre et al., 2016).
These datasets are retrieved by using part of the Siphon (https://github.com/Unidata/siphon) library from UNIDATA.
Concerning model reanalysis, PyRTlib currently considers the ECMWF 5th-generation Reanalysis (ERA5) as accessible from
the Climate Data Store Application Program Interface (CDS API) (https://github.com/ecmwf/cdsapi) service. The following
subsections describe how the above datasets can be accessed through PyRTlib.





## 3.1 Radiosondes

Balloon-borne measurements from radiosondes provide high resolution accurate profiles of temperature, humidity and wind from the altitude of the launching site up to the altitude where the balloon bursts (~ 30 km for a successful launch). This information is an important piece of the global observing system, and it is widely used in atmospheric research and related services, such as operational meteorology, air quality forecast, climatology, NWP validation and data assimilation, and finally the calibration and validation of remote sensing observations. The Wyoming Upper Air Archive from the University of Wyoming consists of radiosonde balloons from more than 628 globally distributed stations over the world. The data are available at synoptic hours (00-06-12-18 UTC) starting from 1973. The available variables are latitude, longitude and elevation of each launching station, and the atmospheric profiles of pressure, geopotential height, temperature, dew point temperature, frost point temperature, relative humidity, relative humidity with respect to ice, water vapour mixing ratio, wind direction, wind speed, potential temperature, equivalent potential temperature, and virtual potential temperature. The vertical resolution varies from tens of meters in the lower layers to hundreds of meters near the tropopause, changing according to the site and weather conditions. Listing 1 shows the code to retrieve and plot data measured by one radiosonde launched at 12 UTC 22 April 2021 from the station named LIRE (Pratica di Mare, Italy), leading to the graphic output in Figure 1. Data from any other station available on the Wyoming Upper Air Archive can be accessed knowing the station name or number that can be found through their web interface (https://weather.uwyo.edu/upperair/sounding.html).

```
import matplotlib.pyplot as plt
from datetime import datetime

from pyrtlib.apiwebservices import WyomingUpperAir
from pyrtlib.utils import to_kelvin
from datetime import datetime

date = datetime(2021, 4, 22, 12)
station = 'LIRE'
df = WyomingUpperAir.request_data(date, station)
df.temperature = to_kelvin(df.temperature)
df.plot("temperature", 'pressure',
        xlabel="T [K]", ylabel="P [hPa]",
        grid=True, legend=False)
df.plot("rh", 'pressure',
        xlabel="RH [%]", ylabel="P [hPa]",
        grid=True, legend=False)
plt.gca().invert_yaxis()
plt.yscale('log')
```

**Listing 1.** Example code using PyRTlib module to retrieve radiosonde data from the Wyoming Upper Air archive.



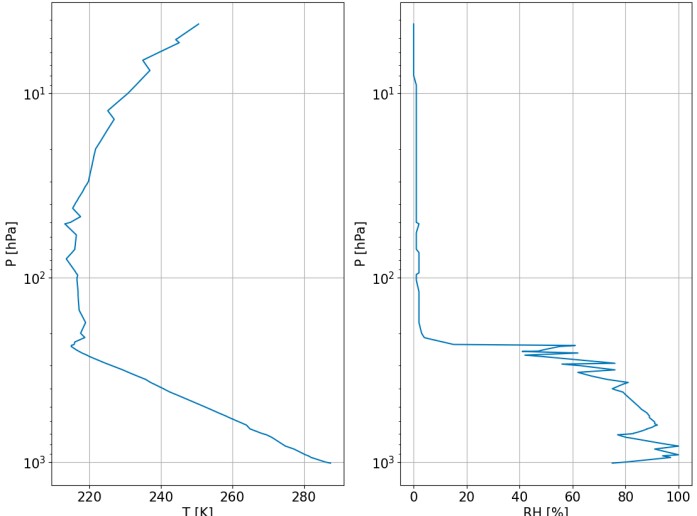

381

**Figure 1.** Graphical output of Listing 1, showing atmospheric profiles measured by the radiosonde launched at 12:00 UTC on
22 April 2021 from the LIRE station (Pratica di Mare, Italy). Left: Temperature profile (K). Right: Relative Humidity (%).

Another well known repository for radiosonde data is the Integrated Global Radiosonde Archive (IGRA), consisting of
radiosonde and pilot balloon observations from more than 2800 stations distributed globally. The earliest data dates back to
1905 and recent data becomes available in near real time from around 800 stations all over the world. The recording period,
temporal and vertical resolution for each station vary over time. Observations are available at standard and variable pressure
levels, fixed and variable height wind levels, surface and tropopause. Variables include time since launch and profiles of
atmospheric pressure, temperature, geopotential height, dew point depression, wind direction and speed at a variable number
of levels, including surface, tropopause, mandatory standard and optional significant levels. The data are released after a quality
assurance algorithm performed by the archiving system, checking for format problems, physically implausible values, internal
inconsistencies among variables, climatological outliers, and temporal and vertical inconsistencies (Durre et al., 2016; 2018).
The IGRA is accessible through NCEI, which also provides access to IGRA station metadata, including information about
changes in the station's location, instrumentation, and observation practices over time, that may be useful for interpreting the
data. Listing 2 shows the code to retrieve and plot data measured by one radiosonde launched at 12 UTC 22 June 2022 from
the station network-id SPM00060018 (Tenerife, Spain), leading to the graphic output in Figure 2. Data from any other station
available on IGRA can be accessed knowing the station network-id that can be found through their web interface
(https://www.ncei.noaa.gov/data/integrated-global-radiosonde-archive/doc/igra2-station-list.txt). Note that PyRTlib provides
tools to convert atmospheric moisture variables to the standard input relative humidity (e.g., in the given example, the function
dewpoint2rh computes relative humidity from dew point depression and physical temperature). PyRTlib then internally
computes water vapour pressure and density from temperature and relative humidity using the Goff-Gratch formulas for





saturation vapour pressure over liquid and ice water, according to a user-specified switch that determines whether the saturation
vapour pressure is calculated over water throughout the profile or over ice when temperature is below a given threshold.

```python
from pyrtlib.apiwebservices import IGRAUpperAir
from pyrtlib.utils import to_kelvin, dewpoint2rh
from datetime import datetime

date = datetime(2022, 6, 22, 12)
station = 'SPM00060018'
df, header = IGRAUpperAir.request_data(date, station)
df = df[df.pressure.notna() &
        df.temperature.notna() &
        df.dewpoint.notna() &
        df.height.notna()]

rh = dewpoint2rh(df.dewpoint, df.temperature).values
df.relative_humidity = rh * 100
df.temperature = to_kelvin(df.temperature)

df.plot("temperature", 'pressure',
        xlabel="T [K]", ylabel="P [hPa]",
        grid=True, legend=False)
df.plot("relative_humidity", 'pressure',
        xlabel="RH [%]", ylabel="P [hPa]",
        grid=True, legend=False)

plt.gca().invert_yaxis()
plt.yscale('log')
```

**Listing 2.** Example code using PyRTlib module to retrieve radiosonde data from the IGRA2 archive.

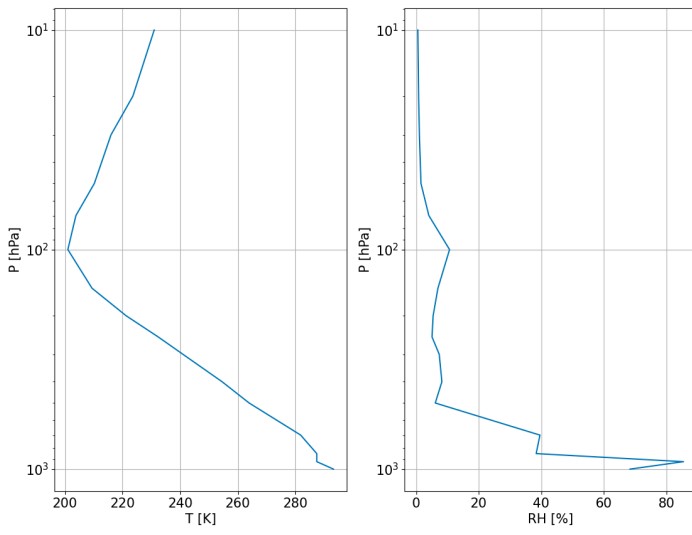

**Figure 2.** Graphical output of Listing 2, showing atmospheric profiles measured by the radiosonde launched at 12:00 UTC on
22 June 2022 from the station SPM00060018 (Tenerife, Spain). Left: Temperature profile (K). Right: Relative Humidity (%).



**3.2 Model reanalysis**

Model reanalysis is an optimal combination of past observations with atmospheric models to provide the most accurate representation of the status of the atmosphere at sub-daily intervals on a regular 3D spatial grid. In short, forecast models and 4D data assimilation systems are used periodically to "reanalyse" archived observations based on variational optimal estimation method. Model reanalysis has substantially evolved during recent decades in generating a consistent time series of multiple climate variables, and are nowadays among the most-used datasets in geophysical sciences. ERA5 is the fifth and latest generation of global climate reanalysis produced by ECMWF, providing hourly data of many atmospheric, land-surface and sea-state parameters together with estimates of uncertainty. ERA5 is based on the most recent and advanced version of the ECMWF Integrated Forecasting System (IFS) model and significantly improved compared to its predecessors (Hersbach et al., 2020). ERA5 is produced and continuously updated by the Copernicus Climate Change Service (C3S) and made available through the Climate Data Store (CDS). ERA5 data are archived on a reduced Gaussian grid, which has quasi-uniform spacing over the globe, at native resolution of 0.28125 degrees (31 km), from the surface up to about 80 km height. Data can be accessed in either GRIB (native) or NetCDF format. PyRTlib implements data retrieval in NetCDF format, which is automatically converted and interpolated from the native grids to a regular latitude/longitude grid (0.125° × 0.125° grid, i.e. ~16 km at the equator) at 37 pressure levels. Hourly estimates of a large number of gridded atmospheric, land and oceanic climate variables are included from 1979 onwards, with a 5-day delay from real time. Among the available variables the following are selected as input for PyRTlib: temperature, relative humidity, specific cloud ice water content, specific cloud liquid water content, ozone mass mixing ratio. Listing 3 shows the code to retrieve ERA5 data from the Copernicus CDS for the nearest grid point to a location in Southern Italy (longitude 16.04°; latitude 39.44°) on 16 May 2023 at 18:00 UTC. Listing 3 also shows tools to convert the native units for cloud water variables (mass mixing ratios, kg/kg) in liquid and ice water density (g/m³), and plots cloud liquid water content (CLWC), cloud ice water content (CIWC), and ozone mass mixing ratio (Figure 3). Data from any other location worldwide from 1979 onwards with a 5-day delay from real time can be accessed by simply providing longitude, latitude, date and hour. Note that to access the ERA5 dataset requires configuring an API key. Step-by-step instructions to create an API key are available at: https://cds.climate.copernicus.eu/api-how-to.

```
import tempfile
from pyrtlib.apiwebservices import ERA5Reanalysis
from pyrtlib.utils import kgkg_to_kgm3

lonlat = (16.04, 39.44)
date = datetime(2023, 5, 16, 18)
nc_file = ERA5Reanalysis.request_data(tempfile.gettempdir(), date, lonlat)

df_era5 = ERA5Reanalysis.read_data(nc_file, lonlat)

total_mass = 1 - df_era5.ciwc.values - df_era5.clwc.values - df_era5.crwc.values - df_era5.cswc.values
denice = df_era5.ciwc.values * (1/total_mass) * \
    kgkg_to_kgm3(df_era5.q.values * (1/total_mass),
                df_era5.p.values, df_era5.t.values) * 1000
denliq = df_era5.clwc.values * (1/total_mass) * \
    kgkg_to_kgm3(df_era5.q.values * (1/total_mass),
```

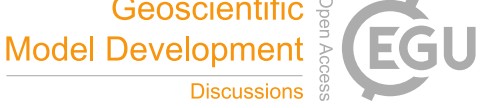

```
                df_era5.p.values, df_era5.t.values) * 1000

df_era5['denice'] = denice
df_era5['denliq'] = denliq

plt.plot(df_era5.denice, df_era5.p, label='CIWC')
plt.plot(df_era5.denliq, df_era5.p, label='CLWC')
plt.xlabel("[$g/m^3$]")
plt.ylabel("P [hPa]")
plt.gca().invert_yaxis()
plt.legend()
plt.grid()

df_era5.plot("o3", 'p',
        xlabel="$O_3$ [$kg/kg$]", ylabel="P [hPa]",
        grid=True, legend=False)

plt.gca().invert_yaxis()
```

**Listing 3.** Example code using PyRTLib module to retrieve atmospheric profiles from the ERA5 Reanalysis.

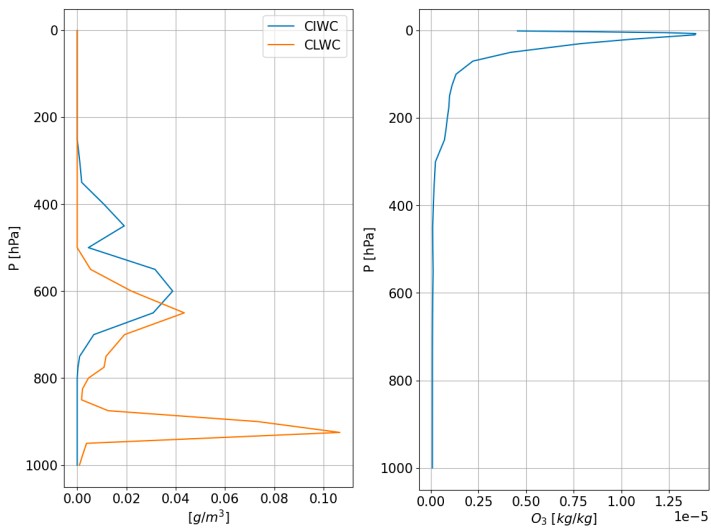

**Figure 3.** Graphical output of Listing 3, showing atmospheric profiles from ERA5 reanalysis for the nearest grid point to a location in Southern Italy (longitude 16.04°; latitude 39.44°) on 16 May 2023 at 18:00 UTC. Left: Cloud liquid water content (CLWC) and cloud ice water content (CIWC). Right: ozone mass mixing ratio.

## 4 Examples of applications

PyRTlib was developed to provide an educational RT software in Python, especially targeting students and younger scientists that mostly use this language for scientific code development. For this reason, PyRTlib was built with additional modules for facilitating the retrieval and management of popular datasets as input, such as radiosonde repository or global reanalysis, as shown in Section 3. This makes PyRTlib a useful end-to-end RT tool for pedagogical purposes, being flexible and interactive with easy access to all kinds of intermediate variables (e.g., absorption, optical depth, opacity, mean radiating temperature). In





addition, PyRTlib was designed to allow easy comparison of MW RT calculations using a set of atmospheric absorption models proposed throughout the last three decades, for any platform (ground-based, airborne, and spaceborne) and observing geometry (zenith, nadir, slant). Finally, PyRTlib implements a general rigorous approach to estimate the uncertainty related to the absorption model (Cimini et al., 2018; Gallucci et al., 2023) and thus it provides $T_B$ calculations with the associated uncertainty estimate, to our knowledge a unique feature in the MW RT scenario. In the following, few examples of applications are given, together with the output figure and the simple code for obtaining it.

**4.1 Simulation of downwelling $T_B$**

As a first example, we propose the simulation of downwelling $T_B$ spectra in a typical MW spectral range. This simple example may turn useful to simulate the measurements from a multi-channel ground-based microwave radiometer, e.g. those widely deployed in atmospheric profiling observatories (Rüfenacht et al., 2021; Shrestha et al., 2022). As input, the user can opt for one of the six climatological atmospheric profiles predefined in PyRTlib (from Anderson et al., 1986: TROPICAL, MIDLAT WINTER, MIDLAT SUMMER, ARCTIC WINTER, ARCTIC SUMMER, US STANDARD) or any of the radiosonde/reanalysis profiles retrieved from the repositories introduced in Section 3. Listing 4 shows the code to compute and plot downwelling $T_B$ spectra at 1-GHz frequency resolution for two typical climatology conditions (tropical and subarctic winter), each at two pointing angles (90° and 30° elevation angle). The graphic output, reported in Figure 4, shows the typical peaks corresponding to resonant absorption of atmospheric gases ($O_2$ at 50-70 and 118 GHz, $H_2O$ at 22.2 and 183 GHz) as well as the non-resonant continuum absorption due to $H_2O$ (monotonically increasing with frequency). The peaks and the continuum show the emission added by the atmospheric gases with respect to the relatively cold emission coming from the outer boundary of the atmosphere (the so called cosmic background). $T_B$ spectra are generally higher for tropical conditions, due to higher atmospheric temperature and humidity with respect to subarctic winter, and for lower elevation angles, due to the slant longer path travelled by radiation throughout the atmosphere.

```
from pyrtlib.tb_spectrum import TbCloudRTE
from pyrtlib.climatology import AtmosphericProfiles as atmp
from pyrtlib.utils import ppmv2gkg, mr2rh

colors = ['#ff0405', '#0404ff', '#0fff0e', '#000000']
atms = [atmp.TROPICAL, atmp.SUBARCTIC_WINTER]
cnt = 0
for atm in atms:
    z, p, _, t, md = atmp.gl_atm(atm)
    gkg = ppmv2gkg(md[:, atmp.H2O], atmp.H2O)
    rh = mr2rh(p, t, gkg)[0] / 100
    frq = np.arange(20, 201, 1)
    ang = np.array([90., 30.])

    for a in ang:
        rte = TbCloudRTE(z, p, t, rh, frq, np.array([a]))
        rte.init_absmdl('R19SD')
        rte.satellite = False
        df = rte.execute()
        df = df.set_index(frq)
```





```
            df.tbtotal.plot(figsize=(12,8), xlabel="Frequency [GHz]", ylabel="Brightness Temperature [K]",
                    label=atmp.atm_profiles()[atm] + ' (' + str(a) + '°)', lw=3, legend=True,
                    color=colors[cnt], grid=True)
        cnt += 1
```

**Listing 4.** Example code using PyRTlib module to calculate downwelling $T_B$ in the 20 to 201 GHz spectral range (1 GHz resolution) using two predefined climatological profiles (Tropical and Subarctic Winter) at 90° and 30° elevation angle.

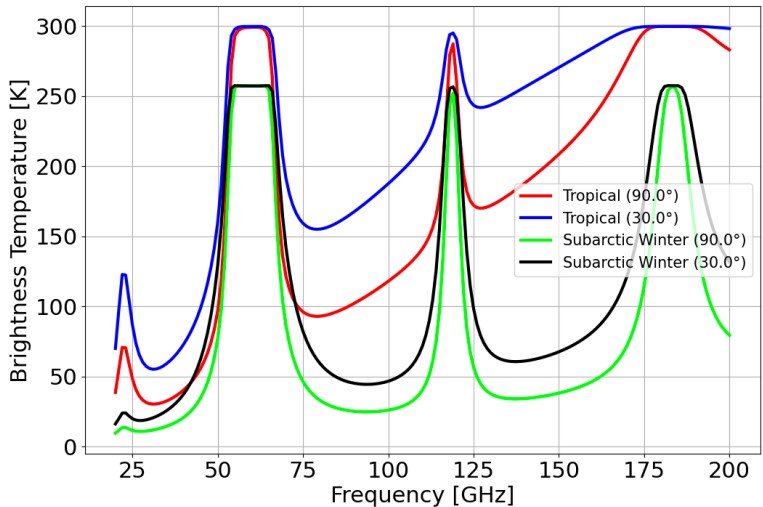

**Figure 4.** Graphical output of Listing 4, showing downwelling $T_B$ spectra computed for two typical climatologies (Tropical and Subarctic Winter) at two elevation angles (90° and 30°).

## 4.2 Simulation of upwelling $T_B$

The second example shows the simulation of upwelling $T_B$ spectra, as those typically sampled by satellite-based MW radiometers (e.g., Moradi et al., 2015). Listing 5 shows the code to compute and plot upwelling $T_B$ spectra at 1-GHz frequency resolution for typical mid-latitude summer climatology conditions. The graphic output, reported in Figure 5, shows that the strong emission features (e.g., at 60-70, 118, 183 GHz) appear flipped with respect to Section 4.1, indicating gas absorption that removes radiation from the emission coming from the relatively warm background Earth's surface. The impact of pointing angle and surface emissivity are shown by varying their values. In particular, 90° pointing angle indicates nadir observations, while 37° indicates typical observing angle of MW imagers (53° from nadir), while 0.9 and 0.45 represent typical high and low emissivity values in the MW spectral region. PyRTlib also allows frequency-dependent surface emissivity to be provided in input.

```
from pyrtlib.tb_spectrum import TbCloudRTE
from pyrtlib.climatology import AtmosphericProfiles as atmp
from pyrtlib.utils import ppmv2gkg, mr2rh

z, p, _, t, md = atmp.gl_atm(atmp.MIDLATITUDE_SUMMER)
gkg = ppmv2gkg(md[:, atmp.H2O], atmp.H2O)
rh = mr2rh(p, t, gkg)[0] / 100
```



```
frq = np.arange(20, 201, 1)
ang = np.array([90., 37.])

colors = ['#ff0405', '#0404ff', '#0fff0e']
cnt = 0
for a in ang:
    rte = TbCloudRTE(z, p, t, rh, frq, np.array([a]))
    rte.init_absmdl('R19SD')
    for e in [0.9, 0.45]:
        if a == 37. and e == 0.90:
            continue
        rte.emissivity = e
        df = rte.execute()
        df = df.set_index(frq)
        df.tbtotal.plot(figsize=(12,8), xlabel="Frequency [GHz]", ylabel="Brightness Temperature [K]",
                        label=atmp.atm_profiles()[atmp.MIDLATITUDE_SUMMER] + ' (E = ' + str(e) + ', angle = ' +
str(a) + '°)',
                        color=colors[cnt], lw=3, legend=True, grid=True)
        cnt += 1
```

**Listing 5.** Example code using PyRTlib module to calculate upwelling T_B in the 20 to 201 GHz spectral range (1 GHz resolution) using a predefined climatological profile (Midlatitude Summer) at 90° and 37° elevation angle with constant surface emissivity at 0.9 and 0.45.

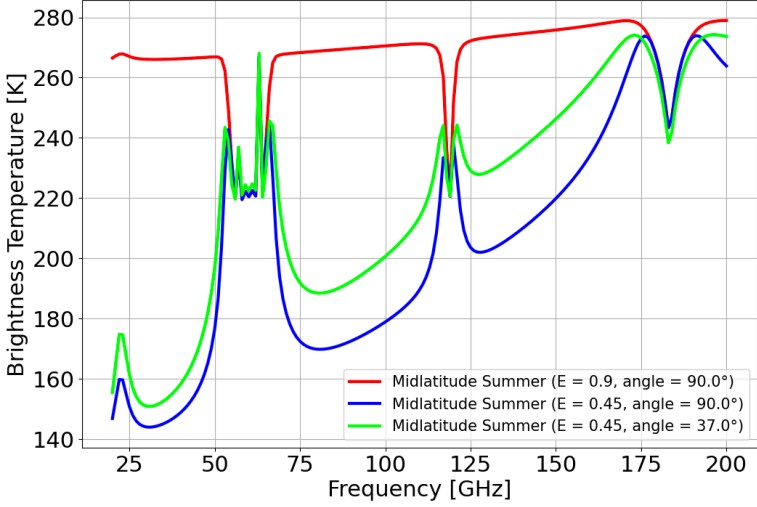

**Figure 5.** Graphical output of Listing 5, showing upwelling T_B spectra computed for a predefined climatological profile (Midlatitude Summer) at 90° and 37° elevation angle with constant surface emissivity at 0.9 and 0.45.

**4.3 Simulation of simultaneous downwelling and upwelling T_B**

Simultaneous observations of downwelling and upwelling T_B are typically performed from airborne scanning instruments that can alternate uplooking and downlooking views (e.g., Fox et al., 2017; Wang et al., 2017). Both views can be simulated by PyRTLib using the same atmospheric profile in input and specifying the altitude of the aircraft and the observing angle. Figure 6 shows the downwelling and upwelling T_B simulated assuming the aircraft at 5 km altitude looking down at nadir and up at





zenith. The input profile comes from a radiosonde launched from Camborne (UK) at 12:00 UTC on 22 July 2021 and retrieved from the Wyoming Upper Air Archive, corresponding to location and period of experimental flights by the Facility for Airborne Atmospheric Measurements (FAAM) BAe-146 aircraft mounting the International Submillimetre Airborne Radiometer (ISMAR, Fox et al., 2017). ISMAR has 17 channels spanning the 118 to 874 GHz range, being developed as an airborne demonstrator for the Ice Cloud Imager (ICI), planned for the second generation of European polar-orbiting satellites (MetOp-SG) to be launched 2024 onwards. Note that PyRTLib functions allow to display and investigate not only $T_B$ but all the intermediate RT variables, such as absorption, optical depth, opacity. For example, Figure 6 shows the atmospheric opacity above and below the aircraft as computed for the uplooking and downlooking views.

**Figure 6.** Top: downwelling and upwelling $T_B$ simulating aircraft observations at respectively zenith and nadir from 5 km altitude (gas absorption model: R22; surface emissivity equal to 1). Bottom: Atmospheric opacity computed for the uplooking





and downlooking views. Input profile from the radiosonde launched from Camborne (UK) on 22 July 2021 at 12:00 UTC and
retrieved from the Wyoming Upper Air Archive. Vertical black lines indicate the ISMAR channel frequencies.

**4.4 Comparison of absorption models**

The PyRTlib package allows $T_B$ simulations with different versions of atmospheric gas absorption models. As mentioned
earlier, the spectroscopy underlying absorption models is continuously updated, following the latest findings from laboratory
and field campaign experiments. Currently, PyRTlib implements absorption routines originally based on the MPM code (Liebe,
1989) with spectroscopy modified throughout the last decades. Specifically, the following versions are readily callable in
PyRTlib (with reference to the paper that motivated the update, where available): R98 (Rosenkranz, 1998); R03 (Tretyakov et
al., 2003), R16, R17 (Rosenkranz, 2017), R19, R19SD (Rosenkranz and Cimini, 2019), R20, R20SD (Makarov et al., 2020),
R21SD (Koshelev et al., 2021), until R22SD (Koshelev et al., 2022). The original Fortran code for most of these absorption
routines by P. W. Rosenkranz are freely accessible through a repository (http://cetemps.aquila.infn.it/mwrnet/lblmrt_ns.html).
In the following, we present an example in which the latest version at the time of writing, R22SD, is compared with R98,
which still represents a widely used model (e.g., Picard et al., 2009). Modifications in R22SD with respect to R98 include:
updated line width at 22 GHz (Payne et al., 2008), updated water vapor continuum coefficients scaled after Turner et al. (2009),
revised $O_2$ mixing coefficients for 50-70 GHz and 118 GHz lines (Makarov et al., 2020), speed-dependent line shape for the
water vapor absorption lines at 22 (Rosenkranz and Cimini, 2019) and 183 GHz (Koshelev et al., 2021), addition of four
submillimeter wave water vapor lines (860, 970, 987, 1097 GHz), and other updated line parameters taken from the most
recent release of the HITRAN database (HITRAN2020). To test two gas absorption versions, a simple observation minus
simulation (O-S) approach can be used, exploiting MW ground-based remote sensing and balloon-borne sounding
measurements from the US Department of Energy (DOE) Atmospheric Radiation Measurement (ARM) program
(https://arm.gov), which can be freely accessed from the ARM data center (https://adc.arm.gov). In fact, ARM deploys a
network of ground-based MW radiometers (MWR) across its observatory sites (Cadeddu et al., 2013; Cadeddu & Liljegren,
2018). These instruments measure downwelling $T_B$ at selected frequency channels under all-sky conditions. From the same
sites, ARM regularly launches radiosondes; the ARM balloon-borne sounding system (BBSS) products provide profiles of in
situ measurements of atmospheric pressure, temperature, humidity, as well as wind speed and direction (Holdridge, 2020),
which can be given in input to PyRTlib to simulate zenith downwelling $T_B$. A dataset of colocated and nearly simultaneous
MWR observations and RT simulations can be then used to test and validate simulations from different absorption models.
Such a dataset was retrieved from the deployment of the ARM mobile facility in Highland Center, Cape Cod (MA, USA)
during the two-column aerosol project (TCAP) in 2012 (Titos et al., 2014). The observations are the $T_B$ measured by a MWR
profiler (MWRP). The MWRP product provides measurements of $T_B$ at 12 frequency channels in the range from 22 to 58 GHz.
Frequencies between 22 and 52 GHz are mostly sensitive to atmospheric water vapor and cloud liquid, while frequencies
between 51 and 60 GHz are sensitive to atmospheric temperature through the absorption of atmospheric oxygen. Simulations
at the same frequency channels are computed from the 4-daily radiosondes (at 05:30, 11:30, 17:30, and 23:30 UTC) launched





during TCAP and processed by PyRTlib. To avoid spatial-temporal uncertainty of clouds, the comparison is made in clear-sky
conditions, applying a cloud screening to both radiosonde and MWRP data. Clear-sky conditions were selected using a relative
humidity threshold, specifically rejecting radiosondes with at least four pressure levels with relative humidity higher than 95%
(Clain et al., 2015). Furthermore, an observation-based screening was applied, removing data for which 1-h standard deviation
of the $T_B$ at 30 GHz was larger than 0.5 K, indicating possible obstructions or cloud contamination (De Angelis et al., 2017).
From a total of 592 radiosonde, these two screenings leave 149 match-ups for the analysis. Simulated and observed datasets
were compared by selecting and averaging MWRP observations falling within -45 minute/+1 hour from each radiosonde
launch time, so as to reduce the temporal-spatial mismatch with respect to the radiosonde measurements. The result of the
comparison of R98 and R22 models versus the observed dataset is shown in Figure 7 and 8.

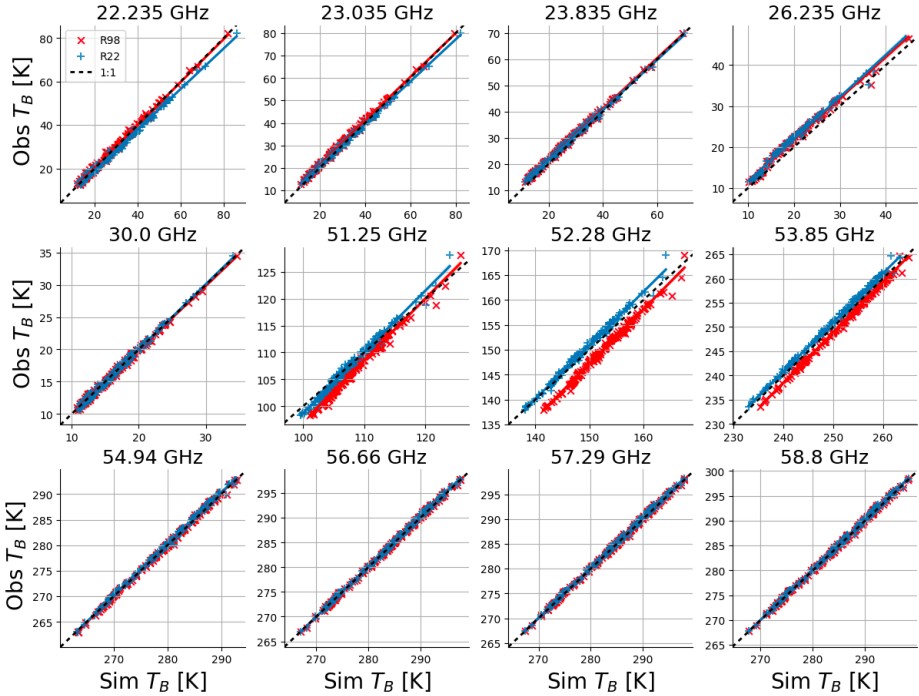

**Figure 7.** Scatter plots of downwelling $T_B$ as observed by a MWRP (y-axis) and simulated with PyRTLib (x-axis) using two
versions of the gas absorption model (R98 in red x and R22 in blue +). Each panel shows one MWRP channel. Markers show
153 MWRP-radiosonde match-ups in clear sky selected from a 6-month period during the TCAP campaign. MWRP and
radiosonde data were retrieved from the ARM data center (Cadeddu and Gibler, 2012; Keeler and Burk, 2012).



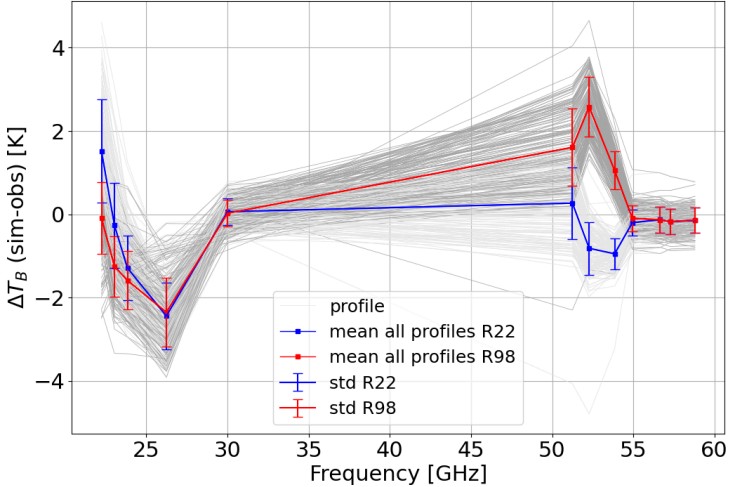


**Figure 8.** Spectrum of simulations minus observations statistics from the 153 MWRP-radiosonde match-ups in clear sky dataset in Figure 7. Simulations computed using R98 are shown in red, while using R22 in blue. Dots and bars indicate respectively the mean value and the standard deviation over the whole dataset.


Figure 7 clearly shows that RT simulations with both absorption models follow nicely observed $T_B$ over the whole range of
variations for all MWRP channels, although larger differences are evident at 51-54 GHz for R98. Bias of the same order of
magnitude for the 51–54 GHz channels were previously reported for the R98 model employing MWR of different types and
manufacturers (e.g., Löhnert and Maier, 2012; De Angelis et al., 2017). De Angelis et al. (2017) attribute these to a combination
of systematic uncertainties stemming from inaccurate instrument bandpass characterization, instrument calibration, and
absorption model. Since then, two major updates have been proposed for the $O_2$ spectroscopic parameters in this range (mainly
mixing coefficients) from laboratory experiments (Tretyakov et al., 2005; Makarov et al., 2020), the latest of which is
implemented in R22. Figure 8 reports mean and standard deviation of the simulation minus observation differences, which
indicate better performances for R22 with respect to R98 in modelling $T_B$ for channels along the low-frequency wing of the
$O_2$ absorption complex, confirming recent independent results (Belikovich et al., 2021; 2022). Unexpectedly, Figure 8 also
indicates differences ~2 K for R22 at 22.2 and 26.235 GHz channels, which will be discussed in next Section. PyRTlib also
allows quantifying the impact on $T_B$ of the most recent set of $O_2$ spectroscopic parameters (Makarov et al., 2020) with respect
to the previous one (Tretyakov et al., 2005). In fact, two absorption model versions implemented in PyRTLib, namely R19
and R22, only differ by this aspect in the 50-70 GHz range, and thus the $T_B$ impact is simply evaluated by computing $T_B$ with
these two versions and taking the difference. To make it general, we evaluate the impact by processing the set of 83 diverse
atmospheric profiles commonly used to train RTTOV (Saunders et al, 2018). This profile set was carefully chosen from more
than 100 million profiles to represent a realistic range of possible diverse atmospheric conditions (Matricardi, 2008) and it is
openly available through the Numerical Weather Prediction Satellite Application Facility (NWPSAF) (https://nwp-
saf.eumetsat.int/site/software/atmospheric-profile-data/). Figure 9 shows the differences between the two models for the



downwelling and upwelling $T_B$ simulated at 50 MHz resolution from the 83 diverse profiles, together with the mean difference
and standard deviation (std). For downwelling $T_B$, the updated $O_2$ mixing coefficients proposed by Makarov et al. (2020)
decrease $T_B$ with respect to the previous ones (Tretyakov et al., 2005) by up to 4-5 K on average, peaking at 53 and 66 GHz,
with an estimated global variability ~1 K (1-sigma). For upwelling $T_B$, the situation is opposite, with a decrease up to -0.6 K
with 0.2 K std, although it depends on the assumed surface emissivity (set to 1 in this example).

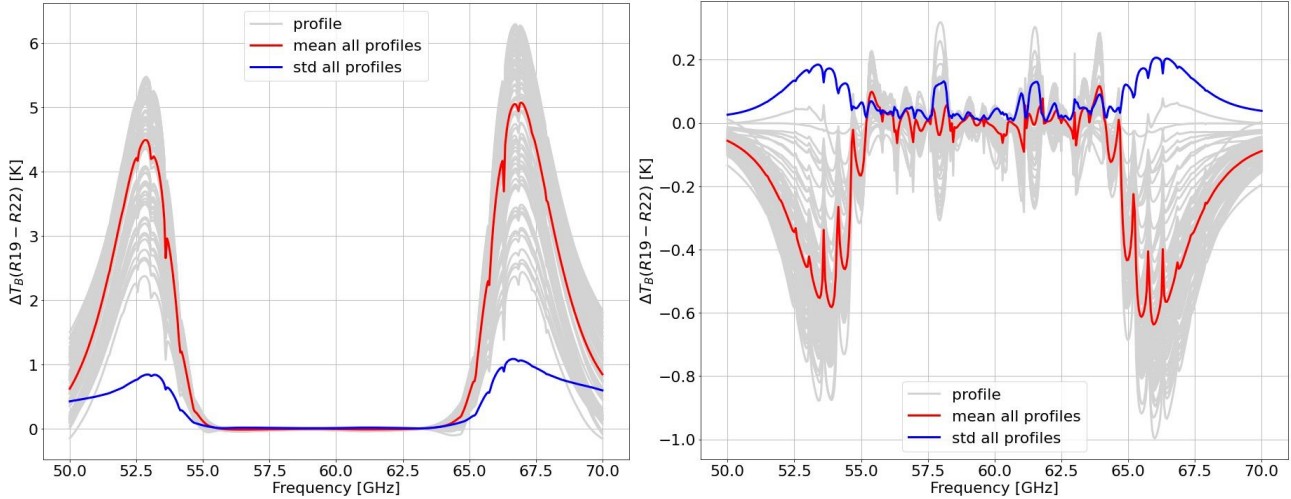


**Figure 9.** Differences of downwelling (left) and upwelling (right) $T_B$ computed with R19 and R22 absorption models at 50
MHz resolution for the set of NWPSAF 83 diverse profiles. Red and blue lines indicate mean difference and standard deviation
of the dataset. Upwelling simulations assume constant surface emissivity equal to 1.

### 4.5 Absorption model uncertainty

RT calculations depend on absorption models and the underlying spectroscopic parameters. The values of these parameters
are determined through theoretical assumptions or analysis of laboratory or field data, and thus are inherently affected by
uncertainty. The uncertainty affecting the spectroscopic parameters contributes to the uncertainty of the absorption, which
affects the RT calculations, and in turn the retrieval of atmospheric variables from remotely sensed observations (Cimini et al.,
2018). PyRTlib allows to compute the sensitivity of RT calculations to the uncertainty of various spectroscopic parameters,
defined as the $T_B$ difference ($\mathbf{\Delta}T_B$) obtained by perturbing the value of spectroscopic parameter by its uncertainty. Figure 10
reports the sensitivity of zenith downwelling $T_B$ to two water vapor absorption spectroscopic parameters, namely the line
intensity ($S$) at 22.2 GHz and the foreign continuum coefficient ($C_f$), showing consistency with results in Cimini et al. (2018,
Figure 2). In addition to the uncertainty of individual parameters, the correlation between the uncertainty on various parameters
must also be taken into account, and therefore it is necessary to calculate the complete uncertainty covariance matrix
(Rosenkranz, 2005). A general and rigorous approach to estimate the uncertainty covariance matrix for MW absorption models
has been outlined (Rosenkranz et al., 2018) and applied to the simulations of downwelling (Cimini et al., 2018) and upwelling



(Gallucci et al., 2023) radiation. PyRTLib inherits this development and provides tools to compute $T_B$ together with the
associated uncertainty estimate. One example is shown in Figure 11. Here, zenith downwelling $T_B$ is computed from one
radiosonde from the TCAP dataset, together with the associated uncertainty estimate $\sigma(T_B)$ and compared with the nearly
simultaneous measurements from the colocated MWRP with its typical calibration uncertainty (Cadeddu & Liljegren, 2018).
Figure 11 shows that for this single case the observation minus simulation differences fit within the overlap of instrumental
calibration uncertainty and absorption model uncertainty at 3-sigma, and for some channels, at 1-sigma level. The only channel
that is nearly off is the 26.235 GHz. This happens also for other radiosondes and absorption models, as evident in Figure 8,
suggesting that the calibration of this particular channel is questionable and may need recalibration service.

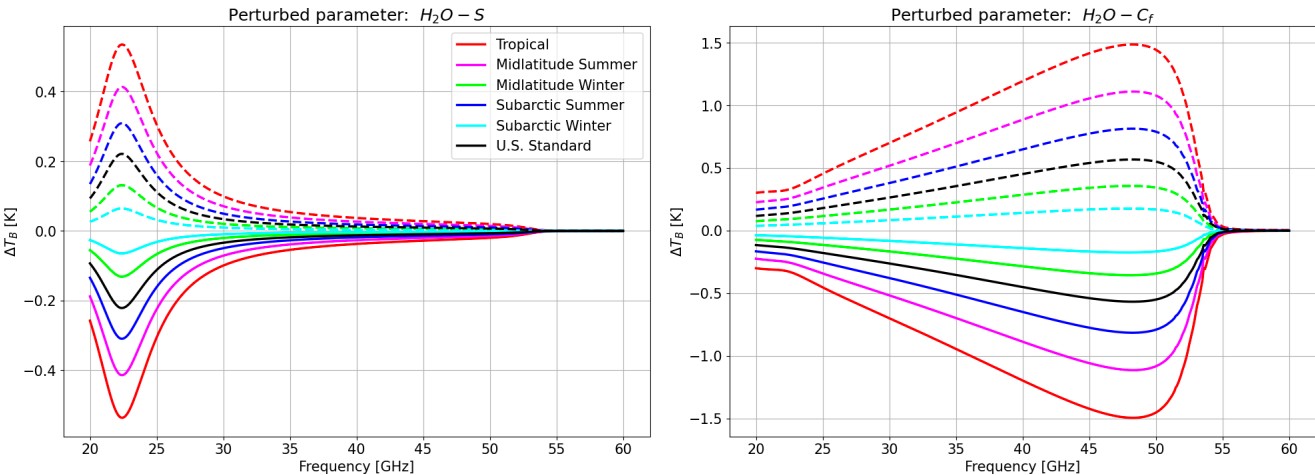

**Figure 10.** Sensitivity of zenith downwelling $T_B$ to water vapor absorption parameters, computed at 0.1 GHz spectral
resolution. Left: line intensity ($S$) at 22 GHz. Right: foreign continuum coefficient ($C_f$). Solid lines correspond to negative
perturbation (value $-$ uncertainty), while dashed lines correspond to positive perturbation (value $+$ uncertainty). Different
colors indicate six typical climatological conditions. Adapted from Cimini et al., 2018.





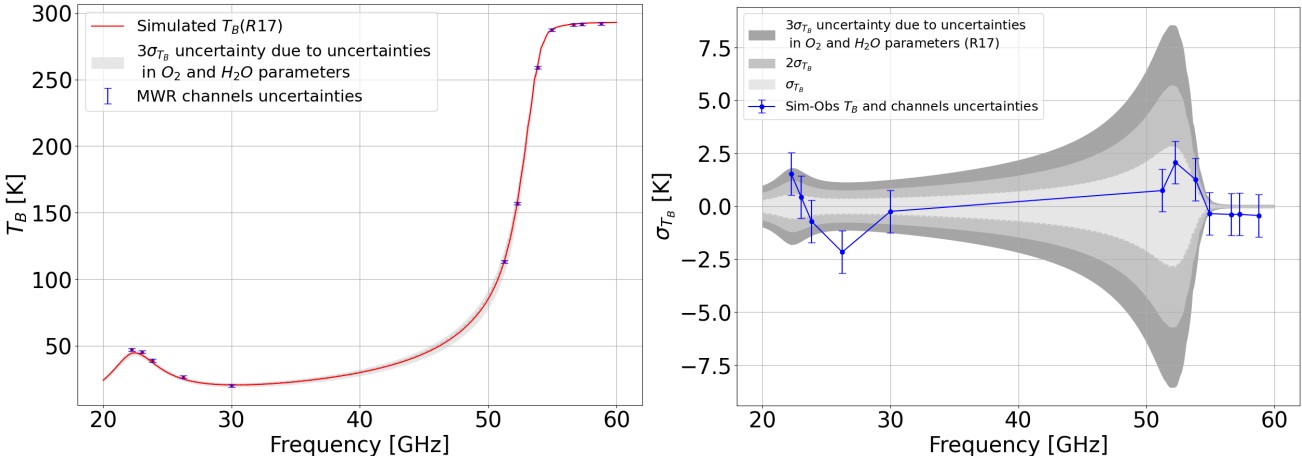

**Figure 11.** Left: Zenith downwelling $T_B$ spectrum (red line) computed from one radiosonde launched from the ARM mobile facility during TCAP, Cape Cod, MA, USA (2012-08-26 16:58 UTC). Blue points and bars indicate the nearly simultaneous measurements from the colocated MWRP at 12 frequency channels and their calibration uncertainty. Right: Simulations minus observations at the 12 channels (blue points) with the instrumental calibration uncertainty (blue bars), together with the estimated uncertainty for zenith downwelling $T_B$ ($\sigma(T_B)$) due to the uncertainty in absorption model spectroscopic parameters (at 1-, 2-, and 3-sigma levels).

## 5 Summary and future developments

This paper presents PyRTlib, a Python library for non-scattering atmospheric microwave radiative transfer computations. The intention for PyRTlib is to provide a user-friendly tool for computing down and up-welling brightness temperatures and related quantities (e.g., atmospheric absorption, optical depth, opacity) in Python, a flexible language that nowadays represents the most used for scientific software development, especially by students and early career scientists. Within its limits, mainly non scattering and 1-D geometry, PyRTlib allows simulating observations from ground-based, airborne, and satellite microwave sensors in clear sky and in cloudy conditions (under Rayleigh approximation). Clearly, the intention for PyRTlib is not to be a competitor for other radiative transfer codes that excel for computational efficiency (RTTOV, CRTM), flexibility (ARTS), modularity (ARTS, Py4CAtS), and applicability (ARTS, PAMTRA). Nevertheless, despite the speed limitations and the omission of important aspects of RT (e.g., spherical geometry and particle scattering), we believe PyRTlib is attractive as an educational tool because of the flexibility and ease of use, providing a quick interface to popular repositories of atmospheric profiles from radiosondes and model reanalysis. PyRTlib also allows peculiar investigations such as absorption model comparison and validation against observations (e.g., Section 4.4) and the estimation of brightness temperature uncertainty due to atmospheric absorption model (e.g., Section 4.5). In addition, PyRTlib could be used as a module for other Python codes that need atmospheric radiative transfer, e.g. the Snow Microwave Radiative Transfer model (SMRT, Picard et al. 2018). Future developments include the implementation of (*i*) new absorption models (e.g. R23 came out at the time of submission),



(*ii*) sensor-oriented calculations considering channels' spectral response functions, (*iii*) uncertainty estimate for higher frequency brightness temperature calculations, as recently investigated (Gallucci et al., 2023), (*iv*) additional tools for extrapolating the input profiles (e.g., Annex 3 of ITU-R P.835-6), (*v*) additional tools for accessing other atmospheric data open repositories to be used as RT calculations input, e.g., the ARM data center and the Global Climate Observing System Reference Upper Air Network (GRUAN, Bodeker et al., 2016).

## 6 Code and data availability

PyRTlib is available as a python package to users under an open-source GPL v3 license and it is free of charge. PyRTlib may be obtained from the github repository https://github.com/SatCloP/pyrtlib or from the Zenodo repository https://doi.org/10.5281/zenodo.8219145. Instructions for installing and running PyRTlib are provided in the PyRTlib User Guide Documentation available at https://satclop.github.io/pyrtlib. The user documentation is rich in content and includes a large number of examples on how to run and configure PyRTlib. The python package also includes scripts and test suite to verify the installation and Jupyter Notebook examples for running the PyRTlib modules to be easily performed in your work environment. PyRTlib is designed for multiplatform systems (UNIX/Linux, MacOS, Windows) and can be installed on any computer supporting Python3.6 (or higher) to work properly.

## 7 Competing interests

The contact author has declared that none of the authors has any competing interests.

## 8 Acknowledgements

The authors acknowledge the support of EUMETSAT through the ISMAR study (contract EUM/CO/20/4600002477/VM) and the VICIRS study (contract EUM/CO/22/4600002714/FDA). The authors also acknowledge the support of ESA through the REFDAT4ESAMWR project (contract 4000134771/21/NL/MG/mkn). PyRTlib development was also stimulated through the COST Action CA18235 PROBE, supported by COST (European Cooperation in Science and Technology, https://www.cost.eu/). The authors acknowledge the advice of Phil W. Rosenkranz (MIT, retired) and Stuart Fox (MetOffice) throughout the software development.

*Author contributions*. SL, DC, and DG designed the research, contributed to data processing and analysis, and wrote the original manuscript. SL and DC developed the software with support from all co-authors. FR contributed to validation data analysis. All the co-authors helped to revise the manuscript.





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
