# Peer review of "PyRTlib: an educational Python-based library for non-scattering atmospheric microwave radiative transfer computations"

_Geoscientific Model Development, 2023_

## Referee Comment (RC1)

**General Comments**

This study appears to be an introduction to a new microwave radiative transfer package written in Python. The development of this package was motivated by the increasingly larger community of Python users. The authors expect that this package will be used for education and incorporated into other Python programs for radiative transfer computations. The paper includes (1) a comprehensive survey of currently available microwave radiative transfer models, (2) a concise description of the basic formulas in microwave radiative transfer computations, and (3) a number of examples of using this newly developed radiative transfer code. Although I am not young enough to choose Python as my primary programming language at present, I appreciate the effort authors put on developing this tool and I think this study fits the scope of *Geoscientific Model Development*. Overall, the paper is well written and easy to read. All of my comments below are straightforward to address and can be regarded as minor.

**Specific Comments**

Lines 473-475. Section 4.2. Figure 5. Yes, for the 183 GHz water vapor absorption band, the three curves in Fig. 5 appear flipped with respect to those in Fig. 4. However, for the 50-70 and 118 GHz $O_2$ absorption bands, only the red curve (surface emissivity = 0.9) in Fig. 5 appears flipped; the green and blue curves (surface emissivity = 0.45) in Fig. 5 do not. It would be helpful if the authors provide an explanation of this feature. Another interesting feature shown in Fig. 5 is that in the 50-70, 118, and 183 GHz gas absorption spectral regions, all the three curves in Fig. 5 are close to each other. It would be wonderful if the authors can also provide an explanation of the insensitivity of simulated upwelling $T_B$ to surface emissivity and sensor viewing angle in these spectral regions.

Lines 497-498. Section 4.3. Figure 6. The legend of Fig. 6b is not very clear. What does "$\tau$ (wet+dry)" stand for? It would be better to introduce this notation in the figure caption. In addition, the authors present the gases that account for the absorption bands in the spectral range between 20 and 201 GHz shown in Figs. 4 and 5. The readers may also be interested in the gases that cause the absorption bands between 118 and 874 GHz shown in Fig. 6.

Lines 597-640. Figure 11 is a good example, but does it have something to do with the uncertainty covariance matrix estimation approach said to be incorporated into the PyRTLib?

**Technical Corrections**

Line 252. The integral in Eq. (5) is from $s_{i-1}$ to $s_i$, not from $s_i$ to $s_{i+1}$. The RHS should be
$$B_f(T_{MR})e^{-\tau_f(0,s_i)}\left[1 - e^{-\tau_f(s_i,s_{i-1})}\right].$$

Line 359. Change "Ccenter" to "Center".

Line 466. "elevation angle" or "elevation angles"?

Line 482. "emissivity at" or "emissivity values of"?

Line 485. "elevation angle" or "elevation angles"? "emissivity at" or "emissivity values of"?

Line 519. What are "wave" water vapor lines?

Lines 541 and 548 and 552. Line 541 says 149 match-ups while lines 548 and 552 both say 153 match-ups.

Line 544. "Figure" of "Figures"?

Not sure if other users also meet the problem, but I could not succeed in running the RT package the authors developed on my machine. After the package was installed, I failed in executing the examples provided by the authors with the same following error message:

`ModuleNotFoundError: No module named 'pyrtlib'`

Under the virtual environment where the package was installed, I tried installing the package again using the command: *pip install pyrtlib* but did not change anything. The following message helped me gain some confidence that the package was successfully installed:

`Requirement already satisfied: pyrtlib in xxx`

---

## Author Comment (AC1)

**Specific Comments**

Lines 473-475. Section 4.2. Figure 5. Yes, for the 183 GHz water vapor absorption band, the three curves in Fig. 5 appear flipped with respect to those in Fig. 4. However, for the 50-70 and 118 GHz $O_2$ absorption bands, only the red curve (surface emissivity = 0.9) in Fig. 5 appears flipped; the green and blue curves (surface emissivity = 0.45) in Fig. 5 do not. It would be helpful if the authors provide an explanation of this feature. Another interesting feature shown in Fig. 5 is that in the 50-70, 118, and 183 GHz gas absorption spectral regions, all the three curves in Fig. 5 are close to each other. It would be wonderful if the authors can also provide an explanation of the insensitivity of simulated upwelling $T_B$ to surface emissivity and sensor viewing angle in these spectral regions.

We thank the reviewer for this comment. Indeed, a deeper explanation was needed for Figure 5. Following the reviewer's comment, we modified the text as follows:

"The graphic output is reported in Figure 5, where the impact of pointing angle and surface emissivity is shown by varying their values. In particular, 90° pointing angle indicates nadir observations, while 37° indicates typical observing angle of MW imagers (53° from nadir), while 0.9 and 0.45 represent typical high and low emissivity values in the MW spectral region. Figure 5 shows that if the emissivity is relatively high (e.g., 0.9), the spectrum resembles that of a warm black-body emission at ~270 K (except where strong atmospheric absorption occurs, e.g., 60, 118, and 183 GHz). Conversely, if the emissivity is relatively low (e.g., 0.45), the background is relatively cold and the atmospheric emission features stick out, similarly to Figure 4. However, near the center of strong emission features (e.g., 60, 118, and 183 GHz) $T_B$ appears flipped with respect to Figure 4, indicating gas absorption that removes radiation from the emission coming from the relatively warm background. It is notable that in those regions $T_B$ nearly overlap for the three emissivity and angle conditions; this is because the atmospheric opacity is so high to make $T_B$ saturate within a short distance, thus becoming insensitive to surface emission and observing angle (i.e., path length)."

Lines 497-498. Section 4.3. Figure 6. The legend of Fig. 6b is not very clear. What does "$\tau$ (wet+dry)" stand for? It would be better to introduce this notation in the figure caption. In addition, the authors present the gases that account for the absorption bands in the spectral range between 20 and 201 GHz shown in Figs. 4 and 5. The readers may also be interested in the gases that cause the absorption bands between 118 and 874 GHz shown in Fig. 6.

With "$\tau$ (wet+dry)" we intend the total opacity ($\tau$) arising from water vapour (wet) and the "dry" air (i.e., the two highest-concentration gases $N_2$ and $O_2$). But we agree this is unclear. So, we modified the figure legend and caption as follows:

**Legend:**
– Atmospheric opacity due to $H_2O$, $O_2$, and $N_2$ above aircraft altitude (5 km)
– Atmospheric opacity due to $H_2O$, $O_2$, and $N_2$ below aircraft altitude (5 km)

**Figure 6.** Top: downwelling and upwelling $T_B$ simulating aircraft observations at respectively zenith and nadir from 5 km altitude (gas absorption model: R22; surface emissivity equal to 1). Bottom: Atmospheric opacity ($\tau$) corresponding to dominant gases ($H_2O$, $O_2$, and $N_2$) computed for the uplooking and downlooking views. All the features correspond to $H_2O$ absorption but the following, due to $O_2$: 118, 368, 424, 487, 715, 773, 834 GHz. Input profile from the radiosonde launched from Camborne (UK) on 22 July 2021 at

12:00 UTC and retrieved from the Wyoming Upper Air Archive. Vertical black lines indicate the ISMAR channel frequencies.

Lines 597-604. Figure 11 is a good example, but does it have something to do with the uncertainty covariance matrix estimation approach said to be incorporated into the PyRTLib?

Correct. The grey shadings correspond to (respectively darker to lighter) 3-sigma, 2-sigma, 1-sigma uncertainty obtained by propagating the spectroscopic parameter uncertainty covariance matrix through the radiative transfer Jacobians (i.e., sensitivity of $T_B$ to spectroscopic parameter). We modified the text as follows to make it clearer:

"One example is shown in Figure 11. It shows ground-based zenith radiometric measurements from the MWRP with its typical calibration uncertainty (Cadeddu & Liljegren, 2018) compared with zenith downwelling $T_B$ computed processing one radiosonde from the TCAP dataset, together with the associated uncertainty estimate $\sigma(T_B)$. $\sigma(T_B)$ is computed within PyRTLib by propagating the spectroscopic parameter uncertainty through the radiative transfer. Calling $\boldsymbol{Cov}(p)$ the parameter uncertainty covariance matrix (as in Cimini et al., 2018), $\boldsymbol{K_p}$ the sensitivity of $T_B$ to spectroscopic parameter (Jacobian), $\sigma(T_B)$ is computed as

$$\sigma(T_B) = diag\big(\boldsymbol{Cov}(T_B)\big) = diag(\boldsymbol{K_p} \cdot \boldsymbol{Cov}(p) \cdot \boldsymbol{K_p^T})$$

where $^T$ indicates the matrix transpose."

**Technical Corrections**

Line 252. The integral in Eq. (5) is from $s_{i-1}$ to $s_i$, not from $s_i$ to $s_{i+1}$. The RHS should be…

Correct. Eq. (5) has been modified accordingly. Thanks for spotting the typo.

Line 359. Change "Ccenter" to "Center".
Line 466. "elevation angle" or "elevation angles"?
Line 482. "emissivity at" or "emissivity values of"?
Line 485. "elevation angle" or "elevation angles"? "emissivity at" or "emissivity values of"?
Line 544. "Figure" of "Figures"?

Agreed. Thanks for spotting the typos.

Line 519. What are "wave" water vapor lines?

The hyphen was missing, sorry. Corrected with "submillimeter-wave". Thanks for spotting the typo.

Lines 541 and 548 and 552. Line 541 says 149 match-ups while lines 548 and 552 both say 153 match-ups.

After both screenings, 149 matchups are left. Thanks for spotting the inconsistency.

Not sure if other users also meet the problem, but I could not succeed in running the RT package the authors developed on my machine. After the package was installed, I failed in executing the examples provided by the authors with the same following error message:

```
ModuleNotFoundError: No module named 'pyrtlib'
```

Under the virtual environment where the package was installed, I tried installing the package again using the command: *pip install pyrtlib* but did not change anything. The following message helped me gain some confidence that the package was successfully installed:

```
Requirement already satisfied: pyrtlib in xxx
```

Thanks for spotting this issue. The "ModuleNotFoundError" could be due to different python installation on the machine. Maybe multiple python versions coexist on the same machine? PyRTlib must be ran through the same python version it was used for its installation. Also, Python v3.7 or higher is required to work with PyRTlib.

Nonetheless, we have now released an upgrade (PyRTlib v1.0.3) which may be installed via the *pip install pyrtlib -U* command. Please have a look.

---

## Author Comment (AC2)

Thank you for considering Py4CAtS in the list of popular RT codes.

Please note that the latest Py4CAtS version incorporates continuum-induced absorption and simple single scattering. It also models aerosol optical depth using simple power law (Angstrom coefficient) and Rayleigh extinction.

Furthermore, the current version incorporates advanced line profiles which take into account effects such as the speed-dependence of pressure broadening, collisional narrowing as well as line mixing in the description of the spectral absorption.

Please also consider to mention that Py4CAtS is available as a Python wheel file from its homepage which allows for an easy installation via the Python package installer pip.

References:

- https://doi.org/10.1016/j.jqsrt.2020.107385 (available at arxiv.org number 2010.09804)

- https://doi.org/10.1016/j.jqsrt.2016.08.009 (available at elib.dlr.de number 106199)

- see documentation in tarball file

Thank you for this short comment. We added the following sentence and references to the revised manuscript to synthetize the new Py4CAtS features you mentioned:

"More recent version of Py4CAtS incorporates continuum-induced absorption, simple single scattering, and modelling of aerosol optical depth, speed-dependence of pressure broadening, including line-mixing (Schreier, 2017; Schreier and Hochstaffl, 2021)."

Reference added:

Schreier, F., Computational aspects of speed-dependent Voigt profiles, Journal of Quantitative Spectroscopy and Radiative Transfer, 187, 44-53, https://doi.org/10.1016/j.jqsrt.2016.08.009, 2017

Schreier, F., Hochstaffl, P., Computational aspects of speed-dependent Voigt and Rautian profiles, Journal of Quantitative Spectroscopy and Radiative Transfer, 258, 107385, https://doi.org/10.1016/j.jqsrt.2020.107385, 2021

---

## Author Comment (AC3)

I thank the authors for including PAMTRA in the list of publically available models. However, I must comment on some points concerning the model's description.

Thanks for your interest in our work. We addressed all the comments as specified below.

PAMTRA includes a variety of gas absorption models where the default selection is the one of Rosenkranz (2015, https://doi.org/10.1109/TGRS.2014.2339015) with various corrections after Turner (2009, https://doi.org/10.1109/TGRS.2009.2022262) and not MPM as mentioned.

We rephrased the sentence to the revised manuscript as follows:

"The user can select several operational modes among scattering and absorption models, within a variety of spectroscopic parameters and databases. The Millimeter-wave Propagation Model (MPM93) developed by Liebe et al. (1993) is used to simulate atmospheric absorption , considering later modifications (e.g., Turner et al., 2009; Rosenkranz, 2015). "

line 115: please correct PAMTRAM to PAMTRA

Thank you for spotting this typo. Amended accordingly.

line 116-118: What is meant by "relative moments" in "regarding radar measurements instead, the active forward model yields Doppler spectra and relative moments, e.g. reflectivity, mean Doppler velocity, skewness, and kurtosis."?

We agree that the sentence could be confusing as is. We modified it in the interest of clarity:

"PAMTRA exploits the passive forward model to compute both upward and downward looking polarized brightness temperatures and radiances for the passive part. The active part can simulate the full radar Doppler spectrum and its higher moments (mean Doppler velocity, skewness, kurtosis)."

line 121-122: Although PyRTlib focuses on non-scattering applications, we would appreciate it if the authors mention that PAMTRA implements the self-similar Rayleigh–Gans approximation (Hogan et al., 2018; https://doi.org/10.1002/qj.2968) for both active and passive applications, a unique feature to our knowledge.

Agreed. We added the following text to the revised version.

"Moreover, PAMTRA implements the self-similar Rayleigh–Gans approximation (SSRGA) for both active and passive applications (Hogan et al., 2017)."

Reference added:
Hogan, R. J., Honeyager, R., Tyynelä, J., and Kneifel, S.: Calculating the Millimetre-Wave Scattering Phase Function of Snowflakes Using the Self-Similar Rayleigh–Gans Approximation, Q. J. Roy. Meteor. Soc., 143, 834–844, https://doi.org/10.1002/qj.2968, 2017.

---

## Author Comment (AC4)

This paper documents a new, Python-based radiative transfer model. This is a welcome addition to the radiation modelling toolkit that is indispensable for weather/climate modelling and remote sensing applications. I would recommend the authors address the following issues, mostly minor, in revision.

A main comment is that as the model is advertised for education, the model package and the paper can be configured to present more diverse and illuminating examples. Currently it's focused/limited to radiance (brightness temperature). Instead of (or in addition to) the input processing, which the paper has talked much about, it would be of more pedagogic value to have examples of computation and diagnosis of different quantities, such as Jacobians, optical depth, weighting function, etc. One heuristic case to showcase the ability of the model may be the logarithmic dependence of monochromatic radiance (Huang & Bani, https://doi.org/10.1002/2014JD022466), which would involve radiance simulation and differencing to verify the phenomenon (log dependence) and involve optical depth and weighting function to understand/explain the phenomenon.

We thank the reviewer for this comment. It's true that the examples in Section 4 present brightness temperatures calculations, so to show the ability to provide the simulations to compare with observations from ground (Section 4.1), satellite (4.2), and airplane (4.3). However, Section 4.3 also shows the computed atmospheric opacity for both uplooking and downlooking views from 5km altitude (Figure 6). Other examples focus on absorption model analysis (4.4), and the sensitivity to spectroscopic parameters and the associated uncertainty (4.5). In addition, Section 6 draws the reader/user to the official documentation, where more examples are given (https://satclop.github.io/pyrtlib/en/main/examples/index.html). To acknowledge the reviewer's suggestion, we added one example to the library, where the logarithmic dependence of monochromatic radiance indicated by Huang & Bani (2014) is reported for 183 GHz. The calculation of weighting functions is not currently available, but it is planned among the future developments and thus was added to the end of Section 5.

Eq 2-5: better to formulate and explain the equations for a slant path, to disclose more fully the parameters that need to be set for the model to run. It would also be better to include necessary description for limb view cases, which the model is said to handle too.

Eqs 2-5 are general, as $s$ indicates the position along the propagation direction, which may be vertical as well as slant. However, we modified the section to clarify the simulation geometry. Note that the current implementation does not handle limb view. In the original manuscript, limb view was only mentioned when describing ARTS. We now specify this clearly in the introduction of the revised manuscript.

Eq. 6: this concerns the treatment of path inhomogeneity, which is a very general problem in radiative modelling. Better to have some reference and discussion of the rationale of the weighting choice made, i.e., based on transmission, optical depth, or mass.

Agreed. We added two more equations and the following discussion to the revised manuscript:

"Note that introducing the layer transmittance $\mathbb{T} = e^{-\tau_f(s_{i-1}, s_i)}$ in Eq.(7), it becomes $B_f(T_{MR}) \simeq \frac{B_f(T(s_{i-1})) + B_f(T(s_i))\mathbb{T}}{1 + \mathbb{T}}$. Thus, $B_f(T_{MR})$ is the average brightness temperatures at the layer boundaries, weighted by the layer transmittance, going from $B_f(T(s_{i-1}))$ to $\frac{B_f(T(s_{i-1})) + B_f(T(s_i))}{2}$ as the layer gets from totally opaque ($\mathbb{T} = 0$) to totally transparent ($\mathbb{T} = 1$). Other weighting options, such as the so-called linear-in-tau, are used elsewhere (Saunders et al., 2018)."

Line 277: parameterization of absorption is core to any radiation model, which is probably covered too briefly here. For education in particular, such essence as state (temperature, pressure, etc.) dependences of line strength and broadening should be reviewed for the concepts as well as for related computing modules.

We opted for high level description of the atmospheric absorption, as this is continuously evolving in the open literature. Several options are implemented in PyRTlib and referenced in Section 4, but we feel the details are beyond the scope of the manuscript. Following the reviewer's suggestion, we included essential information in the revised manuscript.

Line 286: this is another place where I find the information is too brief. It would also be good to outline plans for cloud property parameterization and multiple scattering radiative transfer solver.

As above, essential information have been included in the revised manuscript. Plans for PyRTlib development are outlined in Section 5, but the implementation of a multiple scattering solver is not planned for the near future.

The paper can use some editing assistance for English, e.g., Line 56: "allow to" => allows consideration of …; "dishomogeneity" => inhomogeneity

Agreed. Thanks for spotting the typos.

---

## Author Comment (AC5)

This manuscript introduces a new microwave forward model implemented in Python, PyRTlib. The tool has a relatively narrow scope compared to some existing radiative transfer programs and is brought forward as an educational tool. On the other hand, it has some unique features that are of scientific interest.

As a start, I would suggest putting less emphasis on "educational" (is an educational tool even inside the scope of GMD?). If having education as the main aim, it should be shown that the tool has pedagogical advantages, i.e. how it facilitates learning compared to what can be achieved with other software. As far as I can see, educational is here more or less equal to the second aim of being user-friendly (page 7).

There are two possible target audiences: the students and younger scientists, the "newbies" expected to be the users of PyRTlib, or the teachers and advisers of those users (and there could even be old scientists that master Python and would use PyRTlib in their work!). The manuscript in its present form seems to have both audiences in mind but is not meeting the full expectations of any of them. It could be possible to handle both audiences in parallel, but it is probably best to target one of them. The most reasonable choice seems to prioritise teachers, advisers and scientific use. Targeting newbies would likely result in a too low scientific level for a journal article, and they will primarily look for information in the user guide. However, I fully understand that it is not possible, or even preferred, to have a strict separation between "science" and documentation. Here, a pragmatic approach must be allowed. To be useful, the journal article must contain aspects of documentation, but it should be on a relatively high level, such as the general principles of the software's design.

We thank the reviewer for raising these thoughts. We agree that with "educational" we meant to highlight PyRTlib is "user-friendly" and thus it can be used by students and teacher/advisers with a relatively modest learning curve. We use "educational" also to indicate that several examples are provided with documentation to get the user quickly started. This is not unique of PyRTlib, as it is also true for some of the other available RT codes, though not all. In any case, the purpose of PyRTlib is not to compete with other available RT codes, but to provide a user-friendly alternative with somewhat different, although much narrower, features, i.e.: Python native, quick get started, online absorption model comparison, and spectroscopic uncertainty enabled. The intent is to reach and guide new users (either students or teacher/advisers) to engage with RT modeling, not excluding the scientific community that is active in developing atmospheric microwave absorption models.

The choice of units is an example of general principles. The only way to determine the chosen units is if the example codes clarify them. Please add a table or explain if they are found in the user guide. If the units are selected according to some guidelines (using the ones typically found in atmospheric models?), that would be helpful to know. Further, is there a strategy when importing data? Are the units kept from the original data source or converted to standard units? Taking the function IGRAUpperAir as an example, where do I find out what variables it returns and their unit? The example codes show that the user must employ unit conversions. This is error-prone, and I question whether this aligns with a user-friendly tool.

We thank the reviewer for pointing this out. Indeed, units are a very important aspect for the example codes. Although we had defined the units within the source code, we missed to report into the documentation of the module that retrieves public datasets. Units are kept the same as in the original datasets, although conversion is applied to use the dataset as input and to perform the model in the main function. The units needed of input parameters are shown in the function documentation (https://satclop.github.io/pyrtlib/en/main/generated/pyrtlib.tb_spectrum.TbCloudRTE.__init__.html).

For the public datasets available within PyRTlib (IGRA2, Wyoming, ERA5), the units have been added to the documentation, e.g:

Wyoming:
https://satclop.github.io/pyrtlib/en/main/generated/pyrtlib.apiwebservices.WyomingUpperAir.request_data.html

IGRA2:
https://satclop.github.io/pyrtlib/en/main/generated/pyrtlib.apiwebservices.IGRAUpperAir.request_data.html

ERA5:
https://satclop.github.io/pyrtlib/en/main/generated/pyrtlib.apiwebservices.ERA5Reanalysis.read_data.html

It would be helpful to explain the organisation of the documentation. Where do I look for different types of documentation? The existence of a user guide should be mentioned in the introduction and referred to in the text.

Section 6 "Code and data availability" introduces source code and available documentation. Links to the documentation and to the versioning of source code from GitHub repository are indicated. Currently, documentation is available only for the latest PyRTlib version, but a developer and a version tag documentation are planned for the next release of PyRTlib.

Another missing piece of information is how altitude and observation angle are specified. Please note that it needed to specify an altitude for satellite observations if Eq. 7 is applied. It is the radius of the instrument that sets the constant.

Agreed. We modified the text in Section 2.3 as follows:

"This approximation only requires the elevation angle $\theta$ and the profile of the atmospheric thermodynamical status (pressure, temperature, relative humidity, cloud water content) as function of altitude over the Earth surface (h), and it is the default option in PyRTlib. A one-dimension spherical atmosphere can also be considered, which assumes the atmosphere as uniform concentric layers around a smooth spherical Earth. Following Schroeder and Westwater (1991), the ray path for a spherically stratified atmosphere is modelled through the Snell's law for polar coordinates:

$$n\,r\,cos\theta = constant \tag{9}$$

where n is the atmospheric refractive index and r is the radial distance from the center of the Earth to a point on the ray path. All these qualities depend only on height above the surface. The radiance distance r is assumed as the Earth geoid radius ($R_E$) plus the altitude over the Earth surface. Note that the constant in Eq. (9) must be set in one point of the ray path. This is currently set at the lowest available atmospheric level, which imposes the limitation that PyRTlib version 1.0 cannot simulate limb sounding observations."

I suggest using some tables for better clarity. It is mentioned that "intermediate RT variables" can be accessed, but the exact set is unclear. This information would be suitable for a table, then including units of these variables. A table would also help get a quick overview of the absorption models. Here, I recommend including the reference matching the implementation to avoid confusion.

We thank the reviewer for this suggestion. We added tables in Section 4 to summarize output and all the intermediate variables as follows:

**Table 4.1:** Output variables from PyRTlib.

| Variable | Description | Units |
|----------|-------------|-------|

| | | |
|---|---|---|
| tbtotal | brightness temperature includes cosmic background | K |
| tbatm | atmospheric brightness temperature, no cosmic | K |
| tmr | mean radiating temperature of the atmosphere | K |
| tmrcld | mean radiating temperature of the lowest cloud layer | K |
| taudry | dry air absorption integrated over each ray path | Np |
| tauwet | water vapor absorption integrated over each ray path | Np |
| tauliq | cloud liquid absorption integrated over each ray path | Np |
| tauice | cloud ice absorption integrated over each ray path | Np |

**Table 4.2:** List of all the intermediate variables accessible from PyRTlib.

| Variable | Description | Units |
|---|---|---|
| taulaydry | dry air absorption integrated over each ray path | Np |
| taulaywet | water vapor absorption integrated over each ray path | Np |
| taulayliq | cloud liquid absorption integrated over each ray path | Np |
| taulayice | cloud ice absorption integrated over each ray path | Np |
| srho | water vapor density integrated along each ray path | cm |
| swet | wet refractivity integrated along each ray path | cm |
| sdry | dry refractivity integrated along each ray path | cm |
| sliq | cloud ice density integrated along each ray | cm |
| sice | cloud liquid density integrated along each ray | cm |

Also, the PyRTlib documentation, introduced in Section 6, provides info on all the intermediate output variables at the following link:
https://satclop.github.io/pyrtlib/en/main/generated/pyrtlib.tb_spectrum.TbCloudRTE.execute.html

The results in Figure 7 give confidence in PyRTlib, at least for the two absorption models used. Has there been any comparison between PyRTlib and other forward models? Or internally, comparing the different absorption models? This provides no validation against reality but is the fastest way to detect possible bugs.

It's true that Figure 7 gives confidence in PyRTlib, although the purpose was to show that PyRTlib can be used to monitor MWR observations and to evaluate different absorption models. Several tests have been performed before releasing the software using the available absorption models in PyRTlib. Most of the tests were performed against the original code from NOAA (in Fortran) and later translations (in Matlab). One test was also performed against the results from ARTS using the same absorption model (Rosenkranz, 2021), as shown in the following figure (courtesy of Stuart Fox, extracted from final report of EUMETSAT ISMAR study). The absorption differences between the two implementations are quite small (less than 0.05%) and are attributed to different assumptions in the radiative transfer regarding the variation of absorption coefficient across an atmospheric layer.

We added the following text to Section 4.3: "Atmospheric absorption spectra from PyRTLib were compared with the equivalent computed from ARTS using the same absorption model, resulting in differences within 0.05% (Fox et al., 2023), which are attributed to different assumptions in the variation of absorption coefficient across an atmospheric layer."

[Figure]

Figure 49 (from final report of EUMETSAT ISMAR study): Comparison of absorption coefficients at 3 atmospheric levels from ARTS and CNR using PWR21 spectroscopy. The left-hand column shows the absorption coefficients for each species, and the right-hand column shows the percentage difference between the ARTS and CNR values.

Section 5 outlines extensions. This brings up the issue of the software version. The manuscript must be linked to a specific version. This likely requires a plan to "versionify" the software and document changes between each version.

Correct, we missed version number in Section 5, although it was reported in introduction (Line 200). Thanks for spotting it, we modified the text as follows:

"This paper presents PyRTlib version 1.0, a Python library…"

As mentioned above, PyRTlib source code is stored on a GitHub repository, a version control system tool that keeps track of all the code changes. Furthermore, the PyRTlib repository applies several actions every time the code is modified to test the quality of the code and the installation process and to ensure that PyRTlib is working properly.

Exploring the modelling uncertainty is novel and highly useful and should be better covered by the manuscript.

Agreed. The following information, as well as additional text to better cover the uncertainty modelling, was added to the revised manuscript, referring to the original papers for further details.

Some open questions here:

- Is there an uncertainty estimate for exactly all parameters affecting absorption cross-sections? If not, what parameters are covered?

  The absorption model uncertainty is based on Cimini et al, 2018 (https://doi.org/10.5194/acp-18-15231-2018), where a sensitivity test to parameter uncertainty was performed to identify a reduced set of dominant contributions. This lead to a reduced set of 111 parameters.

- Is it possible to extract these uncertainties and their covariance?

  Yes, the 111x111 uncertainty covariance matrix from Cimini et al, 2018 is accessible through the PyRTlib code (as well as supplement files of the original paper: https://doi.org/10.5194/acp-18-15231-2018).

- How are the parameter uncertainties mapped to Tb uncertainty? Just perturbations or a more advanced approach? Can the full covariance matrix of Tb uncertainty (the information required for OEM retrievals) be obtained?

  Yes, the full covariance matrix of Tb uncertainty is provided in output (e.g., Figure 9 of Cimini et al. 2018). This is obtained by propagating the 111x111 uncertainty covariance matrix through the RT Jacobians (computed by brute force).

I have not tried to install and run PyRTlib. Neither have I cared about minor language issues in detail.

Various comments:

- Why is radiative transfer written with capital R and T in the title? Even if alluding to PyRTlib, this causes more confusion than help.

  Agreed. We used capital letters to recall those in PyRTlib name, but we concur it could be confusing. Capital letters have been changed to lowercase.

- The review of forward models is nice but makes the Introduction long. Place in an appendix?

  We see the point, but prefer to leave the review of forward models in the introduction to introduce all the features that PyRTlib provides and does not provide. Hope this is OK.

- Line 152: AMSUTRAN is only used for the microwave region, not all fast parameterisations.

  Correct. Thanks for spotting this out. We change the text as follows (underlined text is newly added/changed):

"The core of RTTOV is a fast parameterisation of layer optical depths due to gas and liquid water absorption. Profiles of layer-to-space transmittances computed by the line-by-line code AMSUTRAN (Turner et al., 2019) are the basis for the training of the fast parameterisation in the microwave region."

- Line 156: Geer et al (2017) not the best reference here; better is gmd.copernicus.org/articles/14/7497/2021/gmd-14-7497-2021.html

  Thanks for this suggestion. Citation has been modified accordingly.

- Line 194: Not all observation geometries are handled by a plane-parallel model; limb sounding can not be represented.

  Correct. We added the following to the end of Line 194 (underlined text is newly added/changed):

  "(zenith, nadir, slant), except for limb sounding geometry (not currently implemented in PyRTlib);"

- Line 231: As Eq 2 is written, s is the distance from the observation point, not the propagation distance.

  Agreed. Text modified as to: "s indicates the distance from the observations point along the line-of-sight"

- Eq 4: Not correct to use = here. Consider this also for Eq 5.

  Agreed. Eq. 4 has been modified accordingly. For Eq. 5 we added:

  $$B_f(T_{MR}) = \frac{\int_a^b B_f\big(T(s)\big)\, \alpha_f(s)\, e^{-\tau_f(0,s)} ds}{e^{-\tau_f(0,a)}\big[1 - e^{-\tau_f(a,b)}\big]}$$

- Line 268: The value 99% is irrelevant (with this thinking, nitrogen should be the dominating absorber).

  Of course, gas concentration itself does not explain its absorption. In fact, we also specify that "account for most of the gas absorption in the MW spectrum." So, we prefer to keep it unchanged to make it clear that no relevant gas (in terms of concentration and absorption) is neglected.

- Line 291: It is not very clear what has actually been implemented.

  Three models for liquid water absorption are implemented and can be chosen alternatively by the user: one is from Liebe et al. (1993), which updates earlier versions (Liebe et al. 1991), and the other from Rosenkranz (2015). We clarified this in the revised manuscript as described in the next answer below.

- Lines 291-292: Why is this model optional? This implies that the other models are mandatory. Is this model just applied at freezing temperatures?

  The model from Rosenkranz (2015) is optional in the sense that it can be chosen alternatively to the default one (Liebe et al. 1993). It is applied to any temperature as it was developed to be

applicable up to 60°C, but specifically recommended for low temperatures. We rephrased the manuscript as follows for the sake of clarity:

"Another model, as in Rosenkranz (2015), is also implemented for the liquid cloud absorption. This model was developed to be applicable up to 60°C, but it is specifically recommended for temperatures as low as -25°C for modeling the absorption of supercooled liquid water particles. Therefore, currently three models for liquid cloud absorption are available within PyRTlib and can be alternatively selected by the user."

- Eq. 7: Does including refraction for a plane-parallel model make sense? Is it not more important to include the Earth's curvature than refraction? Is including Eq. 7 actually an improvement when applied in a plane-parallel model?

   The plane-parallel approximation is the default option in PyRTlib. However, when the refraction is optionally activated, the ray path is modelled assuming a spherically stratified atmosphere (i.e., 1D atmosphere), as specified in the original manuscript (line 324), which does consider the Earth's curvature. We added text to clarify this aspect in Section 2.3 of the revised manuscript.

- Eqs 8-12: Why this level of detail here? Sticks out compared to other parts. And refraction is of relatively minor concern for the model.

   We concur Eqs. 8-12 are detailed. As mentioned in the acknowledgements, the development of PyRTlib was partially supported by ESA through a project focusing on atmospheric propagation. In this context, the definition of dry and wet refractivity terms may be ambiguous, and thus we prefer to state explicitly the equations implemented as default option.

- Sec 3.1: Are there any issues with quality and missing data when reading from the Wyoming archive?

   For all public datasets available in PyRTlib, we added a simple quality check to control whether the profile is suitable to be used into the radiative transfer equation (e.g., monotonic pressure decrease). Currently this is handled by issuing a warning to the user, although we plan to improve the quality check and error handling in future releases.

- Around here, it becomes clear that the radiative transfer part takes RH as input. Is this a good choice? For freezing temperatures, RH can also be defined with respect to ice. There is a lack of agreement between parameterisations of RH with respect to liquid for freezing temperatures. A user can think that if she/he imports data in RH all is fine, but there is an uncertainty in the mapping from RH to absolute humidity due to two points raised above.

   Correct. RH was chosen as the code was originally developed to process radiosonde data, which only provides RH. The user can choose whether to use RH defined with respect to ice or liquid at freezing temperatures through the "Tconvert" argument (see documentation https://satclop.github.io/pyrtlib/en/main/generated/pyrtlib.utils.mr2rh.html). Whether input, "Tconvert" is used as threshold temperature below which saturation water pressure is calculated over ice instead of liquid water.

- Example codes: One or two example codes are motivated to exemplify how the program is used in practice. The extra examples contribute relatively little (basic things such as the import of packages and plotting get repeated, taking unnecessary space). For the rest, refer to the user documentation.

We thank the reviewer for this comment. The repeated packages import has been removed and we also removed unnecessary code from listing examples, while maintaining the original objective of making the code reproducible and obtaining the same results reported in the figures.

- Line 556: "Nicely" is vague, and not really true (there are apparent deviations from the 1-1 line).

  Agreed. We modified the sentence as follows: "Figure 7 shows that RT simulations with both absorption models tend to align with observed TB over the whole range of variations for all MWRP channels,..."

- Fig. 7: PyRTlib does not apply any bandpass. Can this not be relevant for this comparison?

  Correct, PyRTlib does not apply bandpass. This is on our list for future improvements. Although monochromatic vs band-averaged simulations may differ appreciably, especially along the sharp wing of the $O_2$ complex, we believe bandpass is not the cause of the features in Figure 7. Indeed, similar differences were already noted between bandpass simulations with Rosenkranz 1998 and ground-based MWR observations at 51-54 GHz from independent instruments in different environments (De Angelis et al, 2017; https://doi.org/10.5194/amt-10-3947-2017). This is briefly discussed at lines 556-563 of the original manuscript (now at lines 752-759).

The manuscript is relevant for GMD but requires revision before publication. The text needs better to introduce the actual forward model and its documentation. On the other hand, the background and overall motivation for developing PyRTlib could be shortened (not to extend the length of the manuscript). Some of my comments question design choices. Take these comments as input for discussion and reflection, and not as a demand for making changes to the code at this point.

All the comments were well received, and we believe they improved the manuscript overall. We thank the reviewer for the useful tips and the valuable time dedicated to it.

---

## Author Response (AR2)

**Response to suggestions technical corrections by Report #1**

The revision and reply addressed my comments. Thanks!

It is very interesting to see in the added test case a nice log scaling of water vapour radiative forcing. As far as I know, no one else has shown this behaviour in the microwave frequency in any paper. This looks to me a meaningful science result to include and discuss in this paper.

It would also be useful to list the other (available) test cases in the paper.

Following the reviewer invitation, we added the following Section 4.6 to the manuscript. In addition, Sections 4 and 6 now provide the link to the gallery of PyRTlib examples, which are continuously upgraded. We thank the reviewer for the insightful advices and the valuable time dedicated to our manuscript.

**4.6 Radiative forcing versus water vapour concentration**

The last example presents an interesting feature of the radiative forcing (i.e., radiance change at the top of the atmosphere) caused by greenhouse gases. It has been demonstrated that such a radiative forcing has a logarithmic dependency on the concentration of some greenhouse gases (e.g., $CO_2$ and $H_2O$), and thus logarithmic scaling of e.g. $CO_2$ radiative forcing are often used (IPCC, 2021). This feature is partially attributed to spectrally averaged absorption that saturates logarithmically with the absorber amount (Huang & Bani Shahabadi, 2014), but it was found valid also for infrared monochromatic radiance calculations (Bani Shahabadi and Huang, 2014). To explain that, Huang & Bani Shahabadi (2014) proposed the emission layer displacement (ELD) model, based on the vertical displacement of the most contributing layers, which effectively resolves the radiance change as proportional to the logarithm of the gas concentration. However, assumptions underlying the ELD model do not hold for low-opacity frequencies (e.g., window region). In particular, Bani Shahabadi and Huang (2014) indicate that the logarithmic scaling is valid for relatively opaque frequencies (optical depth >1), while linear scaling is more appropriate for relatively transparent frequencies (optical depth ≤1). To our knowledge, this has not been verified at microwave frequencies yet, though it can be easily tested with PyRTlib as follows. Considering the standard tropical atmosphere and nadir viewing, brightness temperatures are computed at two frequencies corresponding to relatively weak and strong $H_2O$ absorption lines (i.e. 22.235 and 183.0 GHz). For each frequency, $T_B$ are computed seven times by multiplying the water vapor mixing ratio by the following scaling factors ($SF_{q_{H_2O}}$): 1/8, 1/4, 1/2, 1, 2, 4, 8. Figure 12 shows the brightness temperatures difference ($\Delta T_B$) with respect to the unperturbed tropical profile plotted against the binary logarithm of the scaling factor. The logarithmic relationship between $\Delta T_B$ and water vapor concentration is evident for high atmospheric absorption at 183 GHz (opacity ~6 to 262). Conversely, for the relative weak absorption at 22.2 GHz, the relationship changes from linear to logarithmic as the opacity increases from 0.05 to 1.86, showing a knee at ~1 Np.

[Figure]

**Figure 12.** Bottom: Atmospheric opacity $\tau$ at 22.235 (b) and 183.0 GHz (d) vs. $\log_2$ of water vapor concentration scaling factor ($SF_{q_{H_2O}}$). Top: Corresponding change of zenith upwelling monochromatic $T_B$ ($\Delta T_B$) for relatively low opacity at 22.235 GHz (a) and high opacity at 183.0 GHz (c).

**Added references:**
Bani Shahabadi, M., and Y. Huang, Logarithmic radiative effect of water vapor and spectral kernels, J. Geophys. Res. Atmos., 119, 6000–6008, https://doi.org/10.1002/2014JD021623, 2014.
Huang, Y., and M. Bani Shahabadi, Why logarithmic? A note on the dependence of radiative forcing on gas concentration, J. Geophys. Res. Atmos., 119, 13,683–13,689, https://doi.org/10.1002/ 2014JD022466, 2014.
IPCC, 2021: Climate Change 2021: The Physical Science Basis. Contribution of Working Group I to the Sixth Assessment Report of the Intergovernmental Panel on Climate Change [Masson-Delmotte, V., P. Zhai, A. Pirani, S. L. Connors, C. Péan, S. Berger, N. Caud, Y. Chen, L. Goldfarb, M. I. Gomis, M. Huang, K. Leitzell, E. Lonnoy, J. B. R. Matthews, T. K. Maycock, T. Waterfield, O. Yelekçi, R. Yu and B. Zhou (eds.)]. Cambridge University Press, Cambridge, United Kingdom and New York, NY, USA, 2391 pp. doi:10.1017/9781009157896.